# An Adaptive Kernel Approach to Federated Learning of Heterogeneous Causal Effects

**Thanh Vinh Vo**[1]    **Arnab Bhattacharyya**[1]    **Young Lee**[2]    **Tze-Yun Leong**[1]

[1]School of Computing, National University of Singapore
[2]Roche AG and Harvard University
{votv,arnabb,leongty}@nus.edu.sg

## Abstract

We propose a new causal inference framework to learn causal effects from multiple, decentralized data sources in a federated setting. We introduce an adaptive transfer algorithm that learns the similarities among the data sources by utilizing Random Fourier Features to disentangle the loss function into multiple components, each of which is associated with a data source. The data sources may have different distributions; the causal effects are independently and systematically incorporated. The proposed method estimates the similarities among the sources through transfer coefficients, and hence requiring no prior information about the similarity measures. The heterogeneous causal effects can be estimated with no sharing of the raw training data among the sources, thus minimizing the risk of privacy leak. We also provide minimax lower bounds to assess the quality of the parameters learned from the disparate sources. The proposed method is empirically shown to outperform the baselines on decentralized data sources with dissimilar distributions.

## 1   Introduction

Many important questions posed in the natural and social sciences are causal in nature: *What are the long-term effects of mild Covid-19 infection on lung and brain functions? How is mortality rate influenced by the daily air pollution? How would a welfare policy affect employment rate of a minority group?* Causal inference has been applied in a wide range of domains, including economics (Finkelstein and Hendren 2020), medicine (Henderson et al. 2016; Powers et al. 2018), and social welfare (Gutman et al. 2017). The large amount of experimental and/or observation data needed to accurately estimate the causal effects often resides across different sites. In most cases, the data sources cannot be combined to support centralized processing due to some inherent organizational or policy constraints. For example, in many countries, medical or health records of cancer patients are kept strictly confidential at local hospitals; direct exchange or sharing of the records among hospitals, especially for research purposes, are not allowed (Gostin et al. 2009). The main research question is: How to securely access these diverse data sources to build an effective *global causal effect* estimator, while balancing the risk of breaching data privacy and confidentiality?

Current causal inference approaches (e.g., Shalit et al. 2017; Yao et al. 2018) require the shared data to be put in one place for processing. Current federated learning algorithms (e.g., Sattler et al. 2020; Wang et al. 2020) allow collaborative learning of joint models based on non-independent and identically distributed *(non iid)* data; they cannot, however, directly support causal inference as the different data sources might have *disimilar distributions* that would lead to biased causal effect estimation. For example, the demographic profile and average age for cancer patients from two different hospitals may be drastically different. If the two data sets are combined to support causal inference, one distribution may dominate over the other, leading to biased causal effect estimation.

36th Conference on Neural Information Processing Systems (NeurIPS 2022).

We introduce a new approach to federated causal inference from multiple, decentralized, and dissimilarly distributed data sources. Our contributions are summarized as follows:

- We propose a new federated causal inference algorithm, called CausalRFF [1], based on the structural causal model (SCM) (Pearl 2009a), leveraging the Random Fourier Features (Rahimi and Recht 2007) for federated estimation of causal effects. The Random Fourier Features allow the objective function to be divided into multiple components to support federated training of the model.
- We perform federated causal inference with CausalRFF from data sources with different distributions through the *adaptive kernel functions*; the inference is carried out without sharing raw data among the sources, hence minimizing the risk of privacy leak.
- We provide the minimax lower bounds to explicate the limits of estimation and optimization procedures in our federated causal inference framework.

Our work is an important step toward privacy-preserving causal inference. We explore the possibility of combining CausalRFF with multiparty differential privacy at the end of the paper.

## 2  Related Work

Little work has been done on combining causal inference with federated learning in a privacy preserving manner.

**On causal inference:** The authors Hill (2011); Alaa and van der Schaar (2017, 2018); Shalit et al. (2017); Yoon et al. (2018); Yao et al. (2018); Künzel et al. (2019); Nie and Wager (2020) proposed learning causal effects directly from local data sources; these methods adopt the standard *ignorability assumption* (Rosenbaum and Rubin 1983). Louizos et al. (2017); Madras et al. (2019) adapted the structural causal model (SCM) of Pearl (1995) to estimate the causal effects with the existence of latent confounding variables.

Our work is closely related to and extends the notion of *transportability*, where Pearl and Bareinboim (2011); Bareinboim and Pearl (2016); Lee et al. (2020) and related work formulated and provided theoretical analysis of intervention tools on one population to compute causal effects on another population. Lee et al. (2020) generalized transportability to support identification of causal effects in the target domain from the observational and interventional distributions on subsets of observable variables, forming a foundation for drawing conclusions for observational and experimental data (Tsamardinos et al. 2012; Bareinboim and Pearl 2016). Causal inference from multiple, decentralized, dissimilarly distributed sources that cannot be combined or processed in a central site is not addressed. Recently, Aglietti et al. (2020), conducted randomized experiments on the source to collect data and then estimated a joint model of the interventional data from source population and the observational data from target population. Our work is different in that we do not work with randomized data; we estimate causal effects through transfers using only *observational data*. This corresponds to an important setting in real-life, where only retrospective observational data are available, e.g., Covid-19 related case and intervention records, bank and financial transaction records.

**On federated learning:** Federated learning enables collaboratively learning a shared prediction model while keeping all the training data decentralized at source (McMahan et al. 2017). Some federated learning approaches combine federated stochastic gradient descent (Shokri and Shmatikov 2015) and federated averaging (McMahan et al. 2017) to address regression problems Álvarez et al. (2019); Zhe et al. (2019); de Wolff et al. (2020); Joukov and Kulić (2020) and Hard et al. (2018); Zhao et al. (2018); Sattler et al. (2019); Mohri et al. (2019). Recent federated learning algorithms allow collaborative learning of joint deep neural network models based on non-iid data (Sattler et al. 2020; Wang et al. 2020). All these algorithms, however, do not directly support causal inference as the different data sources might have *dissimilar distributions* that would lead to biased causal effect estimation. Little work has been focused on federated estimation of causal effects. Vo et al. (2022) proposed a Bayesian approach that estimates posterior distributions of causal effects based on Gaussian processes, which does not allow *dissimilar distributions* of the sources. Xiong et al. (2021) estimated average treatment effect (ATE) and average treatment effect on the treated (ATT) and assumed that the *confounders are observed*. Our work, on the other hand, estimate conditional average treatment effect (CATE) (which is also known as individual treatment effect, ITE) and average treatment effect (ATE) under the existence of *latent confounders*. We utilize Random Fourier

---

[1]Source code: `https://github.com/vothanhvinh/CausalRFF`

Features to build an integrative framework of causal inference in a federated setting that allows for *dissimilar data distributions*.

## 3 The Proposed Model

In this section, we first detail the problem formalization. We then present the causal effects of interest and the scheme to estimate them. Lastly, we describe the assumptions and the structural equations.

### 3.1 Problem Description

**Problem setting & notations.** Suppose we have $m$ sources of data, each denoted by $\mathsf{D}^\mathsf{s} = \{(w_i^\mathsf{s}, y_i^\mathsf{s}, \boldsymbol{x}_i^\mathsf{s})\}_{i=1}^{n_\mathsf{s}}$, where $\mathsf{s} \in \mathcal{S} := \{\mathsf{s}_1, \mathsf{s}_2, \ldots, \mathsf{s}_m\}$, and the quantities $w_i^\mathsf{s}$, $y_i^\mathsf{s}$ and $\boldsymbol{x}_i^\mathsf{s}$ are the treatment assignments, observed outcome associated with the treatment, and covariates of individual $i$ in source $\mathsf{s}$, respectively. These data sources $\mathsf{D}^\mathsf{s}$ are located in different locations and their distributions might be completely different. All the sources share the same causal graph as shown in Figure 1, but the data distributions may be different, e.g., $p_{\mathsf{s}_1}(\boldsymbol{x}, w, y) \neq p_{\mathsf{s}_2}(\boldsymbol{x}, w, y)$, where $p_{\mathsf{s}_1}(\cdot)$ and $p_{\mathsf{s}_2}(\cdot)$ denote the two distributions on two sources $\mathsf{s}_1$ and $\mathsf{s}_2$, respectively. Similarly, the marginal and the conditional distributions with respect to these variables can also be different (or similar). The objective is to develop a global causal inference model that satisfies *both* of the following two conditions: **(i)** the causal inference model can be trained in a private setting where the data of each source are not shared to an outsider, and **(ii)** the causal inference model can incorporate data from multiple sources to improve causal effects estimation in each specific source.

**Causal effects of interest.** Given a causal model trained under the aforementioned setting, we are interested in estimating the conditional average treatment effect (CATE)[2] and average treatment effect (ATE). Let $Y$, $W$, $X$ be random variables denoting the outcome, treatment, and proxy variable, respectively. Then, the CATE and ATE are defined as follows (Louizos et al. 2017; Madras et al. 2019)

$$\tau(\boldsymbol{x}) := E\big[Y|\mathrm{do}(W{=}1), X{=}\boldsymbol{x}\big] - E\big[Y|\mathrm{do}(W{=}0), X{=}\boldsymbol{x}\big], \qquad \tau := E[\tau(X)], \qquad (1)$$

where $\mathrm{do}(W{=}w)$ represents that a treatment $w \in \{0, 1\}$ is given to the individual. This definition is followed from Louizos et al. (2017); Madras et al. (2019). Given a set of $n$ *new* individuals whose covariates/observed proxy variables are $\{\boldsymbol{x}_i\}_{i=1}^n$, the CATE and ATE in this sub-population are obtained by $\tau(\boldsymbol{x}_i)$ and $\tau = \sum_{i=1}^n \tau(\boldsymbol{x}_i)/n$.

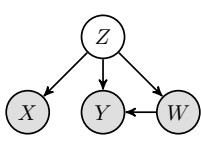

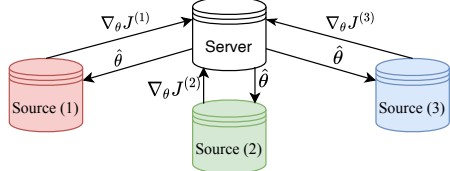

Figure 1: The causal graph with latent confounder $Z$, treatment $W$, outcome $Y$, covariate/proxy variable $X$.

Figure 2: An example of our proposed model with three sources. The objective function $J \simeq J^{(1)} + J^{(2)} + J^{(3)}$ is decomposed to 3 components, each associated with a source.

The central task to estimate CATE and ATE is to find $E[Y \,|\, \mathrm{do}(W = w), X = \boldsymbol{x}]$. Since the data distribution of each source might be different from (or similar to) each other, we use the notation $E[Y|\mathrm{do}(W = w^\mathsf{s}, X = \boldsymbol{x}^\mathsf{s}]$ to denote the expectation of the outcome $Y$ under an intervention on $W$ of an individual in source $\mathsf{s}$. With the existence of the latent confounder $Z$, we can further expand this quantity using *do*-calculus (Pearl 1995). In particular, from the backdoor adjustment formula, we have

$$E\big[Y|\mathrm{do}(W = w^\mathsf{s}), X = \boldsymbol{x}^\mathsf{s}\big] = \int E\big[Y|W = w^\mathsf{s}, Z = \boldsymbol{z}^\mathsf{s}\big] p_\mathsf{s}(\boldsymbol{z}^\mathsf{s}|\boldsymbol{x}^\mathsf{s}) d\boldsymbol{z}^\mathsf{s}. \qquad (2)$$

Eq. (2) shows that the causal effect is identifiable if we can find the conditional distributions $p_\mathsf{s}(y^\mathsf{s}|w^\mathsf{s}, \boldsymbol{z}^\mathsf{s})$ and $p_\mathsf{s}(\boldsymbol{z}^\mathsf{s}|\boldsymbol{x}^\mathsf{s})$ for each source $\mathsf{s}$. The second distribution can be further expanded by $p_\mathsf{s}(\boldsymbol{z}^\mathsf{s}|\boldsymbol{x}^\mathsf{s}) = \sum_{w^\mathsf{s}} \int p_\mathsf{s}(\boldsymbol{z}|\boldsymbol{x}^\mathsf{s}, y_i^\mathsf{s}, w^\mathsf{s}) p_\mathsf{s}(y^\mathsf{s}|\boldsymbol{x}^\mathsf{s}, w^\mathsf{s}) p_\mathsf{s}(w^\mathsf{s}|\boldsymbol{x}^\mathsf{s}) \mathrm{d}y^\mathsf{s}$. Following the forward sampling strategy, the remaining is to find the following distributions

$$p_\mathsf{s}(w^\mathsf{s}|\boldsymbol{x}^\mathsf{s}), \qquad p_\mathsf{s}(y^\mathsf{s}|\boldsymbol{x}^\mathsf{s}, w^\mathsf{s}), \qquad p_\mathsf{s}(\boldsymbol{z}^\mathsf{s}|\boldsymbol{x}^\mathsf{s}, y^\mathsf{s}, w^\mathsf{s}), \qquad p_\mathsf{s}(y^\mathsf{s}|w^\mathsf{s}, \boldsymbol{z}^\mathsf{s}), \qquad (3)$$

---

[2]Also called individual treatment effect (ITE)

and then systematically draw samples from these estimated distributions to obtain the empirical expectation of $Y$ given $\text{do}(W = w^{\text{s}})$ and $X = \boldsymbol{x}^{\text{s}}$.

**Identification.** The CATE and ATE are identifiable if we are able to learn the distributions in Eq. (3), which involve latent confounder $Z$. Louizos et al. (2017) showed that this is possible if $Z$ has a relationship to the observed variables $X$, and there are many cases that it is identifiable such as: $Z$ is categorical and $X$ is a Gaussian mixture model (Anandkumar et al. 2014), $X$ includes three independent views of $Z$ (Goodman 1974; Allman et al. 2009; Anandkumar et al. 2012), $Z$ is a multivariate binary and $X$ are noisy functions of $Z$ (Jernite et al. 2013; Arora et al. 2017), to name a few. Following the works by Louizos et al. (2017); Madras et al. (2019), we use variational inference in the spirit of the variational auto-encoder (VAE) to recover the latent confounders, since it can learn a rich class of latent-variable models, and thus recovering the causal effects. Identification of our work follows closely from the literature, however our main contribution is in the *federated setting* of the model. Please refer to Appendix for the proof of identifiability.

## 3.2 The Causal Graph and Assumptions

Since our method adopts the SCM approach with the causal graph in Figure 1, there are some implicit assumptions that follow from the axioms and properties of SCM: **(A1)** *Consistency*: $W = w \implies Y(w) = Y$, this follows from the axioms of SCM. **(A2)** *No interference*: the treatment on one subject does not affect the outcomes of another one. This is because the outcome has only a *single treatment node* as its parent. **(A3)** *Positivity*: every subject has some positive probability to be assigned to every treatment. These assumptions are standard in any causal inference algorithm. One can find further discussion in Pearl (2009a,b); Morgan and Winship (2015). For our proposed federated setting, we make two additional assumptions as follows:

**(A4)** The individuals in all sources have the same set of *common* covariates.

**(A5)** Any individual does not exist in more than one source.

Assumption **(A4)** has been implicitly shown in our setup since all the sources would share the same causal graph. This is a reasonable assumption as we intend to build a unified model on all of the data sources, e.g., decentralized data in Choudhury et al. (2019); Vaid et al. (2020); Flores et al. (2020) satisfy this assumption for federated learning. Assumption **(A5)** is to ensure that no individuals would dominate the other individuals when training the model. For example, if an individual appears in all of the sources, the trained model would be biased by data of this individual (there is imbalance caused by the use of more data from this particular individual than the others). Hence, this condition would ensure that such bias does not exist. In practice, Assumption **(A5)** sometimes does not hold. To address such a problem, we perform a *pre-training step* to exclude such duplicated individuals. This step would use a one-way hash function to perform a secured matching procedure that identifies duplicated individuals. Details of the pre-training step are presented in Appendix.

## 3.3 The Structural Equations

This section presents how the causal relations are modeled. Since $Z$ is the root node in the causal graph, we model it as a multivariate normal distribution: $Z \sim \mathsf{N}(\boldsymbol{\mu}, \sigma_z^2 \mathbf{I}_{d_z})$ for the all sources. We now detail the structural equations of $Y, W$ and $X$. Let $V$ be a univariate variable that represents a node or a dimension of a node in the causal graph (Figure 1), i.e., $V$ can be $Y$, $W$ or a dimension of $X$. Let $\mathsf{pa}(V)$ be set of $V$'s parent variables in the causal graph, i.e, the nodes with directed edges to $V$. We model the structural equation of $V$ in two cases as follows:

$$\text{if } V \text{ is continuous:} \quad V = f_v(\mathsf{pa}(V)) + \epsilon_v, \quad \text{if } V \text{ is binary:} \quad V = \mathbb{1}[\varphi(f_v(\mathsf{pa}(V))) > \epsilon_v], \quad (4)$$

where $\epsilon_v \sim \mathsf{N}(0, \sigma_v^2)$ for the former case and $\epsilon_v \sim \mathsf{U}[0,1]$ for the latter case, $\varphi(\cdot)$ is the logistic function and $\mathbb{1}(\cdot)$ is the indicator function. The latter case implies that $V$ given $\mathsf{pa}(V)$ follows Bernoulli distribution with $p(V = 1|\mathsf{pa}(V)) = \varphi(f_v(\mathsf{pa}(V)))$. Furthermore, if $W \in \mathsf{pa}(V)$, then we further model

$$f_v(\mathsf{pa}(V)) \;=\; (1 - W)f_{v0}(\mathsf{pa}(V) \setminus \{W\}) \;+\; W f_{v1}(\mathsf{pa}(V) \setminus \{W\}). \quad (5)$$

**Example.** If $Y \in \mathbb{R}$, $W \in \{0, 1\}$ and $X_k \in \mathbb{R}$ ($X_k$ is the $k$–th dimension of $X$), then the structural equations are as follows:

$$Y = (1 - W)f_{y0}(Z) + W f_{y1}(Z) + \epsilon_y, \qquad W = \mathbb{1}[\varphi(f_w(Z)) > \epsilon_w], \qquad X_k = f_{x_k}(Z) + \epsilon_{X_k}.$$

In the subsequent sections, we present how to learn the functions $f_v$ ($v \in \{y0, y1, w, x\}$) in a federated setting and then use them to estimate the causal effects of interest.

# 4 CausalRFF: An Adaptive Federated Inference Algorithm

This section presents a new federated algorithm to learn the distributions in Eq. (3). The central task is to decompose the objective function into multiple components, each associated with a source.

## 4.1 Learning Distributions Involving Latent Confounder

To estimate causal effects, we need to estimate the four quantities detailed in Eq. (3). This section presents how to learn $p_{\mathsf{s}}(\boldsymbol{z}^{\mathsf{s}}|\boldsymbol{x}^{\mathsf{s}}, y^{\mathsf{s}}, w^{\mathsf{s}})$ and $p_{\mathsf{s}}(y^{\mathsf{s}}|w^{\mathsf{s}}, \boldsymbol{z}^{\mathsf{s}})$. Since the marginal likelihood has no analytical form, we learn the above distributions using variational inference which maximizes the evidence lower bound (ELBO)

$$\mathcal{L} = \sum_{\mathsf{s} \in \mathcal{S}} \sum_{i=1}^{n_{\mathsf{s}}} \Big( E_q \big[ \log p_{\mathsf{s}}(y_i^{\mathsf{s}}|w_i^{\mathsf{s}}, \boldsymbol{z}_i^{\mathsf{s}}) + \log p_{\mathsf{s}}(w_i^{\mathsf{s}}|\boldsymbol{z}_i^{\mathsf{s}}) + \log p_{\mathsf{s}}(\boldsymbol{x}_i^{\mathsf{s}}|\boldsymbol{z}_i^{\mathsf{s}}) \big] - \mathrm{KL}[q(\boldsymbol{z}_i^{\mathsf{s}}) \| p(\boldsymbol{z}_i^{\mathsf{s}})] \Big), \quad (6)$$

where $q(\boldsymbol{z}^{\mathsf{s}}) = \mathsf{N}(\boldsymbol{z}^{\mathsf{s}}; f_q(y^{\mathsf{s}}, w^{\mathsf{s}}, \boldsymbol{x}^{\mathsf{s}}), \sigma_q^2 \mathbf{I})$ is the variational posterior distribution. The function $f_q(\cdot)$ is modeled as follows: $f_q(y^{\mathsf{s}}, w^{\mathsf{s}}, \boldsymbol{x}^{\mathsf{s}}) = (1 - w^{\mathsf{s}})f_{q0}(y^{\mathsf{s}}, \boldsymbol{x}^{\mathsf{s}}) + w^{\mathsf{s}} f_{q1}(y^{\mathsf{s}}, \boldsymbol{x}^{\mathsf{s}})$, where $f_{q0}$ and $f_{q1}$ are two functions to be learned. The density functions $p_{\mathsf{s}}(y^{\mathsf{s}}|w^{\mathsf{s}}, \boldsymbol{z}^{\mathsf{s}})$, $p_{\mathsf{s}}(w^{\mathsf{s}}|\boldsymbol{z}^{\mathsf{s}})$ and $p_{\mathsf{s}}(\boldsymbol{x}^{\mathsf{s}}|\boldsymbol{z}^{\mathsf{s}})$ are obtained from the structural equations as described in Section 3.3. Please refer to Appendix for details on derivation of the ELBO.

**Adaptive modeling.** Since the observed data from each source might come from different (or similar) distributions, we would model them separately and adaptively learn their similarities. In particular, we propose a kernel-based approach to learn these distributions. To proceed, we first obtain the empirical loss function $\widehat{\mathcal{L}}$ from negative of the ELBO $\mathcal{L}$ by generating $M$ samples of each latent confounder $Z$ using the reparameterization trick (Kingma and Welling 2013): $\boldsymbol{z}_i^{\mathsf{s}}[l] = f_q(y_i^{\mathsf{s}}, w_i^{\mathsf{s}}, \boldsymbol{x}_i^{\mathsf{s}}) + \sigma_q \epsilon_i^{\mathsf{s}}[l]$, where $\epsilon_i^{\mathsf{s}}[l]$ is drawn from the standard normal distribution. We obtain a complete dataset

$$\widetilde{\mathsf{D}}^{\mathsf{s}} = \bigcup_{l=1}^{M} \big\{ (w_i^{\mathsf{s}}, y_i^{\mathsf{s}}, \boldsymbol{x}_i^{\mathsf{s}}, \boldsymbol{z}_i^{\mathsf{s}}[l]) \big\}_{i=1}^{n_{\mathsf{s}}}, \quad \forall \mathsf{s} \in \mathcal{S}. \quad (7)$$

Using this complete dataset, we minimize the following objective function

$$J = \widehat{\mathcal{L}} + \sum_{c \in \mathcal{A}} R(f_c) \quad (8)$$

with respect to $f_c$, where $\mathcal{A} = \{y0, y1, w, x, q0, q1\}$, and $R(\cdot)$ denotes a regularizer. The minimizer of $J$ would result in the following form of $f_c$

$$f_c(\boldsymbol{u}^{\mathsf{s}}) = \sum_{\mathsf{v} \in \mathcal{S}} \sum_{j=1}^{n_{\mathsf{v}} \times M} \kappa(\boldsymbol{u}^{\mathsf{s}}, \boldsymbol{u}_j^{\mathsf{v}}) \boldsymbol{\alpha}_j^{\mathsf{v}}, \quad (9)$$

where $\boldsymbol{u}_j^{\mathsf{v}}$ is obtained from the $j$–th tuple of the dataset $\tilde{\mathsf{D}}^{\mathsf{v}}$. Details are presented in Appendix. Since data from the sources might come from a completely different (or similar) distribution, we would use an adaptive kernel to measure their similarity. In particular, let $k(\boldsymbol{u}^{\mathsf{s}}, \boldsymbol{u}^{\mathsf{v}})$ be typical kernel function such as squared exponential kernel, rational quadratic kernel, or Matérn kernel. The kernel used in Eq. (9) is as follows: $\kappa(\boldsymbol{u}^{\mathsf{s}}, \boldsymbol{u}^{\mathsf{v}}) = \lambda^{\mathsf{s},\mathsf{v}} k(\boldsymbol{u}^{\mathsf{s}}, \boldsymbol{u}^{\mathsf{v}})$, if $\mathsf{s} \neq \mathsf{v}$; otherwise, $\kappa(\boldsymbol{u}^{\mathsf{s}}, \boldsymbol{u}^{\mathsf{v}}) = k(\boldsymbol{u}^{\mathsf{s}}, \boldsymbol{u}^{\mathsf{v}})$, where $\lambda^{\mathsf{s},\mathsf{v}} \in [0, 1]$ is the adaptive factor and it is learned from the observed data.

**Remark.** Eq. (9) indicates that computing $f_c(\boldsymbol{u}^{\mathsf{s}})$ requires collecting all data points from all sources, and so the objective function in Eq. (8) cannot be optimized in a federated setting. Next, we present a method known as Random Fourier Features to address the problem.

**Random Fourier Features.** We show how to adapt Random Fourier Features (Rahimi and Recht 2007) into our model. Let $k(\boldsymbol{u}, \boldsymbol{u}')$ be any translation-invariant kernel (e.g., squared exponential kernel, rational quadratic kernel, or Matérn kernel). Then, by Bochner's theorem (Wendland 2004, Theorem 6.6), it can be written in the following form:

$$k(\boldsymbol{u}, \boldsymbol{u}') = \int e^{i\boldsymbol{\omega}^{\top}(\boldsymbol{u} - \boldsymbol{u}')} s(\boldsymbol{\omega}) d\boldsymbol{\omega} = \int \cos\big(\boldsymbol{\omega}^{\top}(\boldsymbol{u} - \boldsymbol{u}')\big) s(\boldsymbol{\omega}) d\boldsymbol{\omega}, \quad (10)$$

where $s(\boldsymbol{\omega})$ is a spectral density function associated with the kernel (please refer to Appendix for spectral density of some popular kernels). The last equality follows from the fact that the kernel function is real-valued and symmetric. This type of kernel can be approximated by

$$k(\boldsymbol{u}, \boldsymbol{u}') \simeq B^{-1} \sum_{b=1}^{B} \cos(\boldsymbol{\omega}_b^\top (\boldsymbol{u} - \boldsymbol{u}')) = \phi(\boldsymbol{u})^\top \phi(\boldsymbol{u}'), \qquad \{\boldsymbol{\omega}_b\}_{b=1}^{B} \overset{i.i.d.}{\sim} s(\boldsymbol{\omega}), \qquad (11)$$

where $\phi(\boldsymbol{u}) = B^{-\frac{1}{2}} [\cos(\boldsymbol{\omega}_1^\top \boldsymbol{u}),..., \cos(\boldsymbol{\omega}_B^\top \boldsymbol{u}), \sin(\boldsymbol{\omega}_1^\top \boldsymbol{u}),..., \sin(\boldsymbol{\omega}_B^\top \boldsymbol{u})]^\top$. The last equality follows from the trigonometric identity: $\cos(u - v) = \cos u \cos v + \sin u \sin v$. Substituting the above random Fourier Features into Eq. (9), we obtain

$$f_c(\boldsymbol{u}^{\mathsf{s}}) \simeq \Big(\theta_c^{\mathsf{s}} + \sum_{\mathsf{v} \in \mathcal{S} \setminus \{\mathsf{s}\}} \lambda^{\mathsf{s},\mathsf{v}} \theta_c^{\mathsf{v}}\Big)^\top \phi(\boldsymbol{u}^{\mathsf{s}}), \qquad (12)$$

where $\theta_c^{\mathsf{s}} = \sum_{i=1}^{n_{\mathsf{s}}} \phi(\boldsymbol{u}^{\mathsf{s}}) \boldsymbol{\alpha}_i^{\mathsf{s}}$ and $\lambda^{\mathsf{s},\mathsf{v}}$ (s, v $\in \mathcal{S}$). While optimizing the objective function $J$, instead of learning $\boldsymbol{\alpha}_i^{\mathsf{s}}$, we can directly consider $\theta^{\mathsf{s}}$ as parameter to be optimized. This has been used in several works such as Rahimi and Recht (2007); Chaudhuri et al. (2011); Rajkumar and Agarwal (2012). This approximation allows us to rewrite the objective function $J$ as a summation of local objective functions in each source:

$$J \simeq \sum_{\mathsf{s} \in \mathcal{S}} J^{(\mathsf{s})}, \qquad \text{where } J^{(\mathsf{s})} = \widehat{\mathcal{L}}^{(\mathsf{s})} + m^{-1} \sum_{\mathsf{v} \in \mathcal{S}} \zeta \|\theta^{\mathsf{v}}\|_2^2, \qquad (13)$$

where $\zeta \in \mathbb{R}^+$ is a regularizer factor. Each component $J^{(\mathsf{s})}$ is associated with the source s and it can be computed with the local data in this source. Hence, it enables federated optimization for the objective function $J$. Figure 2 illustrates our proposed federated causal learning algorithm with three sources, where $\theta$ denotes the set of all parameters to be learned including $\theta^{\mathsf{s}}$ and $\lambda^{\mathsf{s},\mathsf{v}}$ from all the sources. The federated learning algorithm can be summarized as follows: First, each source computes the local gradient, $\nabla_\theta J^{(\mathsf{s})}$, using its own data and sends to the server. The server, then, collects these gradients from all sources and subsequently updates the model. Next, the server broadcasts the new model to all the sources.

**Minimax lower bound.** We now compute the minimax lower bound of the proposed model, which gives the rate at which our estimator can converge to the population quantity of interest as the sample size increases. We first state the following result that concerns the last two terms in Eq. (3):

**Lemma 1** (With presence of latent variables). *Let* $\theta = \{\theta_c^{\mathsf{s}} : c \in \{y0, y1, x, w\}, \mathsf{s} \in \mathcal{S}\}$ *and* $\hat{\theta}$ *be its estimate. Let* $y_i^{\mathsf{s}} \in \mathbb{R}$ *and* $\boldsymbol{x}_i^{\mathsf{s}} \in \mathbb{R}^{d_x}$. *Let* $\mathcal{S}_{\setminus \mathsf{s}} = \mathcal{S} \setminus \{\mathsf{s}\}$. *Then,*

$$\inf_{\hat{\theta}} \sup_{P \in \mathcal{P}} \mathbb{E}_P \Big[\|\hat{\theta} - \theta(P)\|_2\Big] \geq \frac{\sqrt{m(d_x + 3)} \log(2\sqrt{m})}{64\sqrt{B} \sum_{\mathsf{s} \in \mathcal{S}} n_{\mathsf{s}} \big(1 + \sum_{\mathsf{v} \in \mathcal{S}_{\setminus \mathsf{s}}} \lambda^{\mathsf{s},\mathsf{v}}\big)^2}. \qquad (14)$$

The LHS of Eq. (14) can be seen as the worst case of the best estimator, whereas the RHS depicts the behavior of the convergence. The bounds do not only depend on the number of samples ($n_{\mathsf{s}}$, training size) of each source but also the adaptive factors $\lambda^{\mathsf{s},\mathsf{v}}$. When the adaptive factors are small, the lower bounds are large since data from a source s are only used to learn its own parameter $\theta^{\mathsf{s}}$. When the adaptive factors are large, the lower bounds are smaller, which suggests that data from a source would help infer parameters associated with the other sources. This bound gives a guarantee on how data from all the sources impact the learned parameters that modulate the two distributions $p_{\mathsf{s}}(\boldsymbol{z}^{\mathsf{s}} | \boldsymbol{x}^{\mathsf{s}}, y^{\mathsf{s}}, w^{\mathsf{s}})$ and $p_{\mathsf{s}}(y^{\mathsf{s}} | w^{\mathsf{s}}, \boldsymbol{z}^{\mathsf{s}})$. The proof of Lemma 1 can be found in Appendix.

### 4.2 Learning Auxiliary Distributions

The previous section has shown how to learn $p_{\mathsf{s}}(\boldsymbol{z}^{\mathsf{s}} | \boldsymbol{x}^{\mathsf{s}}, y^{\mathsf{s}}, w^{\mathsf{s}})$ and $p_{\mathsf{s}}(y^{\mathsf{s}} | w^{\mathsf{s}}, \boldsymbol{z}^{\mathsf{s}})$. To compute treatment effects, we need to learn two more conditional distributions, namely $p_{\mathsf{s}}(w^{\mathsf{s}} | \boldsymbol{x}^{\mathsf{s}})$ and $p_{\mathsf{s}}(y^{\mathsf{s}} | \boldsymbol{x}^{\mathsf{s}}, w^{\mathsf{s}})$. Since all the variables in these two distributions are observed, we estimate them using maximum likelihood estimation. In the following, we present a federated setting to learn $p_{\mathsf{s}}(w^{\mathsf{s}} | \boldsymbol{x}^{\mathsf{s}})$. Similar to the previous section, the objective function here can also be decomposed into $m$ components as follows: $J_w \simeq \sum_{\mathsf{s} \in \mathcal{S}} J_w^{(\mathsf{s})}$, where $J_w^{(\mathsf{s})} = \sum_{i=1}^{n_{\mathsf{s}}} \ell(w_i^{\mathsf{s}}, \varphi(g(\boldsymbol{x}_i^{\mathsf{s}}))) + m^{-1} \sum_{\mathsf{v} \in \mathcal{S}} \zeta_w \|\psi^{\mathsf{v}}\|_2^2$ and $g(\boldsymbol{x}_i^{\mathsf{s}}) = \sum_{\mathsf{v} \in \mathcal{S}} \phi(\boldsymbol{x}_i^{\mathsf{s}})^\top (\psi^{\mathsf{s}} + \gamma^{\mathsf{s},\mathsf{v}} \psi^{\mathsf{v}})$, $\gamma^{\mathsf{s},\mathsf{v}} \in [0, 1]$ is the adaptive factor, $\psi^{\mathsf{s}}$ is the parameter associated with source s, and $\ell(\cdot)$ denotes the cross-entropy loss function since $w_i^{\mathsf{s}}$ is a binary value. The first component of $J_w^{(\mathsf{s})}$ is obtained from the negative log-likelihood. Learning of $p_{\mathsf{s}}(y^{\mathsf{s}} | \boldsymbol{x}^{\mathsf{s}}, w^{\mathsf{s}})$ is

similar. For convenience, in the subsequent analyses, we denote the parameters and adaptive factors of this distribution as $\beta^{\mathsf{s}}$ and $\eta^{\mathsf{s},\mathsf{v}}$, where $\mathsf{s}, \mathsf{v} \in \mathcal{S}$ and $\mathsf{s} \neq \mathsf{v}$. The next lemma shows the minimax lower bound for the first two sets of parameters $\psi$ and $\beta$ in Eq. (3), but this time without involving the latent variables:

**Lemma 2** (Without the presence of latent variables)**.** *Let* $\psi = \{\psi^{\mathsf{s}}\}_{\mathsf{s}=1}^{m}$, $\beta = \{\beta^{\mathsf{s}}\}_{\mathsf{s}=1}^{m}$ *and* $\hat{\psi}$, $\hat{\beta}$ *be their estimates, respectively. Let* $y_i^{\mathsf{s}} \in \mathbb{R}$. *Then,*

**(i)** $$\inf_{\hat{\psi}} \sup_{P \in \mathcal{P}} \mathbb{E}_P \left[ \|\hat{\psi} - \psi(P)\|_2 \right] \geq \frac{m \log(2\sqrt{m})}{256 \sum_{\mathsf{s} \in \mathcal{S}} n_{\mathsf{s}} \left(1 + \sum_{\mathsf{v} \in \mathcal{S}_{\backslash \mathsf{s}}} \gamma^{\mathsf{s},\mathsf{v}}\right)}, \qquad (15)$$

**(ii)** $$\inf_{\hat{\beta}} \sup_{P \in \mathcal{P}} \mathbb{E}_P \left[ \|\hat{\beta} - \beta(P)\|_2 \right] \geq \frac{\sigma}{2^{\frac{9}{2}}} \left( \frac{m \log(2\sqrt{m})}{B \sum_{\mathsf{s} \in \mathcal{S}} n_{\mathsf{s}} \left(1 + \sum_{\mathsf{v} \in \mathcal{S}_{\backslash \mathsf{s}}} \eta^{\mathsf{s},\mathsf{v}}\right)^2} \right)^{1/2}. \qquad (16)$$

The proof of Lemma 2 can be found in Appendix. The bounds presented in Lemma 1 and 2 give helpful information about the number of samples to be observed and the cooperation of multiple sources of data through the transfer factors. Since we used variational inference and maximum likelihood to learn the parameters in our model, these methods give consistent estimation as shown in Kiefer and Wolfowitz (1956); Van der Vaart (2000); Wang and Blei (2019); Yang et al. (2020).

## 4.3  Computing Causal Effects

The key to estimate causal effects in our model is to compute the outcome in Eq. (2). We proceed by drawing samples from the distributions in Eq. (3). Generating samples from the conditional distributions $p_{\mathsf{s}}(w^{\mathsf{s}}|x^{\mathsf{s}})$, $p_{\mathsf{s}}(y^{\mathsf{s}}|x^{\mathsf{s}}, w^{\mathsf{s}})$, and $p_{\mathsf{s}}(y^{\mathsf{s}}|w^{\mathsf{s}}, z^{\mathsf{s}})$ is straightforward since they are readily available as shown in either Section 4.1 or 4.2. There are two options to draw samples from the posterior distribution of confounder $p_{\mathsf{s}}(z^{\mathsf{s}}|x^{\mathsf{s}}, y^{\mathsf{s}}, w^{\mathsf{s}})$. The first one is to draw from its approximation, $q(z^{\mathsf{s}})$, since maximizing the ELBO in Section 4.1 is equivalent to minimizing $\mathrm{KL}(q(z^{\mathsf{s}})\|p_{\mathsf{s}}(z^{\mathsf{s}}|x^{\mathsf{s}}, y^{\mathsf{s}}, w^{\mathsf{s}}))$. As a second option, we note that the exact posterior of confounder can be rewritten as $p_{\mathsf{s}}(z^{\mathsf{s}}|x^{\mathsf{s}}, y^{\mathsf{s}}, w^{\mathsf{s}}) \propto p_{\mathsf{s}}(y^{\mathsf{s}}|z^{\mathsf{s}}, w^{\mathsf{s}})p_{\mathsf{s}}(w^{\mathsf{s}}|z^{\mathsf{s}})p_{\mathsf{s}}(x^{\mathsf{s}}|z^{\mathsf{s}})p(z^{\mathsf{s}})$, whose components on the right hand side are also available in Section 4.1. Thus, we can draw from this distribution using the Metropolis-Hastings (MH) algorithm. Since $Z$ is a multidimensional random variable, the traditional MH algorithm would require a long chain to converge. We overcome this problem by using the MH with independent sampler (Liu 1996) where the proposal distribution is the variational posterior distribution $q(z^{\mathsf{s}})$ learned in Section 4.1. The second approach would give more accurate samples since we select the samples based on exact acceptance probability of the posterior $p_{\mathsf{s}}(z^{\mathsf{s}}|x^{\mathsf{s}}, y^{\mathsf{s}}, w^{\mathsf{s}})$. This would help estimate the CATE given $x_i^{\mathsf{s}}$. The local ATE is the average of CATE of individuals in a source $\mathsf{s}$. These quantities can be estimated in a local source machine. To compute a global ATE, the server would collect all the local ATE in each source and then compute their weighted average. Further details are in Appendix.

## 5  Experiments

**The baselines.** In this section, we first carry out the experiments to examine the performance of CausalRFF against standard baselines such as BART (Hill 2011), TARNet (Shalit et al. 2017), CFR-wass (CFRNet with Wasserstein distance) (Shalit et al. 2017), CFR-mmd (CFRNet with maximum mean discrepancy distance) (Shalit et al. 2017), CEVAE (Louizos et al. 2017), OrthoRF (Oprescu et al. 2019), X-learner (Künzel et al. 2019), R-learner (Nie and Wager 2020), and FedCI (Vo et al. 2022). In contrast to CausalRFF, these methods (except FedCI) do not consider causal inference within a federated setting. We compare our method to these baselines trained in two ways: (**a**) training a global model with the combined data from all the sources, (**b**) using bootstrap aggregating of Breiman (1996) where $m$ models are trained separately on each source data and then averaging the predicted treatment effects based on each trained model. Note that case (**a**) *violates* federated data setting and is only used for comparison purposes. In general, we expect that the performance of CausalRFF to be close to that of the performance of the baselines in case (**a**) when the data distribution of all the sources are the same. In addition, we also show that the performance of CausalRFF is better than that of the baselines in case (**a**) when the data distribution of all the source are different.

**Implementation of the baselines.** The implementation of CEVAE is from Louizos et al. (2017). Implementation of TARNet, CFR-wass, and CFR-mmd are from Shalit et al. (2017). For these

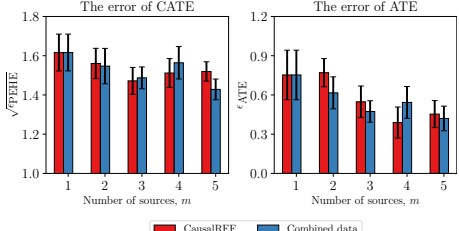

Figure 3: Experimental results on DATA^same.

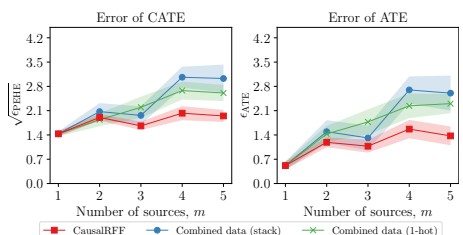

Figure 4: Experimental results on DATA^diff.

Table 1: Out-of-sample errors on DATA^same where top-3 performances are highlighted in bold (lower is better). The dashes (-) in 'ag' (bootstrap aggregating) indicate that the numbers are the same as that of 'cb' (combined data).

| Method | The error of CATE, $\sqrt{\epsilon_{PEHE}}$ | | | The error of ATE, $\epsilon_{ATE}$ | | |
|---|---|---|---|---|---|---|
| | 1 source | 3 sources | 5 sources | 1 source | 3 sources | 5 sources |
| BART$_{ag}$ | - | 3.8±.10 | 3.8±.09 | - | 2.3±.15 | 2.3±.14 |
| X-Learner$_{ag}$ | - | 3.2±.07 | 3.1±.06 | - | 0.6±.11 | 0.5±.13 |
| R-Learner$_{ag}$ | - | 3.5±.17 | 3.9±.46 | - | 1.5±.35 | 2.0±.70 |
| OthoRF$_{ag}$ | - | 5.4±.21 | 4.5±.12 | - | 0.5±.10 | 0.7±.16 |
| TARNet$_{ag}$ | - | 3.9±.04 | 3.4±.03 | - | 2.2±.07 | 2.0±.02 |
| CFR-wass$_{ag}$ | - | 3.0±.05 | 3.6±.02 | - | 2.1±.03 | 1.8±.02 |
| CFR-mmd$_{ag}$ | - | 4.0±.03 | 3.9±.02 | - | 2.3±.03 | 2.0±.01 |
| CEVAE$_{ag}$ | - | **2.9±.04** | **2.5±.04** | - | 0.7±.08 | **0.5±.10** |
| BART$_{cb}$ | **3.7±.12** | 3.2±.07 | 3.1±.03 | 2.1±.20 | 1.0±.18 | 0.6±.13 |
| X-Learner$_{cb}$ | 3.3±.06 | 3.4±.06 | 3.3±.04 | **0.5±.11** | **0.4±.06** | **0.5±.12** |
| R-Learner$_{cb}$ | 4.2±.46 | 3.4±.07 | 3.4±.04 | 2.2±.72 | 0.6±.15 | 0.9±.15 |
| OthoRF$_{cb}$ | 7.6±.29 | 4.3±.10 | 3.7±.07 | 1.4±.30 | **0.4±.12** | **0.5±.10** |
| TARNet$_{cb}$ | 4.2±.07 | 3.8±.03 | 3.5±.02 | 2.2±.13 | 2.1±.06 | 2.1±.03 |
| CFR-wass$_{cb}$ | 4.0±.11 | 3.8±.02 | 3.7±.02 | 2.1±.06 | 2.0±.03 | 1.9±.02 |
| CFR-mmd$_{cb}$ | 3.8±.05 | 3.8±.02 | 3.7±.02 | 2.1±.04 | 2.1±.03 | 2.0±.02 |
| CEVAE$_{cb}$ | **2.5±.03** | **2.4±.03** | **2.4±.03** | **0.5±.08** | **0.3±.06** | **0.3±.06** |
| FedCI | **2.5±.03** | **2.4±.03** | **2.5±.03** | **0.4±.06** | **0.3±.11** | **0.3±.10** |
| CausalRFF | **1.6±.09** | **1.5±.07** | **1.5±.05** | **0.8±.19** | **0.5±.12** | **0.4±.10** |

methods, we use Exponential Linear Unit (ELU) activation function and fine-tune the number of nodes in each hidden later from 10 to 200 with step size of addition by 10. For BART, we use package `BartPy`, which is readily available. For X-learner and R-learner, we use the package `causalml` (Chen et al. 2020). For OrthoRF, we use the package `econml` (Microsoft Research 2019). For FedCI, we use the code from Vo et al. (2022). For all methods, the learning rate is fine-tuned from $10^{-4}$ to $10^{-1}$ with step size of multiplication by 10. Similarly, the regularizer factors are also fine-tuned from $10^{-4}$ to $10^0$ with step size of multiplication by 10. We report two error metrics: $\epsilon_{PEHE}$ (precision in estimation of heterogeneous effects) and $\epsilon_{ATE}$ (absolute error) to compare the methods. We report the mean and standard error over 10 replicates of the data. Further details are presented in Appendix.

### 5.1 Synthetic Data

**Data description.** Obtaining ground truth for evaluating causal inference algorithm is a challenging task. Thus, most of the state-of-the-art methods are evaluated using synthetic or semi-synthetic datasets. In this experiment, the synthetic data is simulated with the following distributions:

$$\boldsymbol{z}_i^s \sim \mathsf{Cat}(\rho), \qquad x_{ij}^s \sim \mathsf{Bern}(\varphi(a_{j0} + (\boldsymbol{z}_i^s)^\top \mathbf{a}_{j1})), \qquad w_i^s \sim \mathsf{Bern}(\varphi(b_0 + (\boldsymbol{z}_i^s)^\top(\mathbf{b}_1 + \Delta))),$$
$$y_i^s(0) \sim \mathsf{N}(\mathsf{sp}(c_0 + (\boldsymbol{z}_i^s)^\top(\mathbf{c}_1 + \Delta)), \sigma_0^2), \qquad y_i^s(1) \sim \mathsf{N}(\mathsf{sp}(d_0 + (\boldsymbol{z}_i^s)^\top(\mathbf{d}_1 + \Delta)), \sigma_1^2),$$

where $\mathsf{Cat}(\cdot)$, $\mathsf{N}(\cdot)$, and $\mathsf{Bern}(\cdot)$ denote the categorical distribution, normal distribution, and Bernoulli distribution, respectively. $\varphi(\cdot)$ denotes the sigmoid function, $\mathsf{sp}(\cdot)$ denotes the softplus function, and $\boldsymbol{x}_i = [x_{i1},...,x_{id_x}]^\top \in \mathbb{R}^{d_x}$ with $d_x = 30$. Herein, we convert $\boldsymbol{z}_i^s$ to a one-hot vector. To simulate data, we randomly set the ground truth parameters as follows: $\rho = [.11, .17, .34, .26, .12]^\top$, $(c_0, d_0) = (0.9, 7.9)$, $(\mathbf{c}_1, \mathbf{d}_1, \mathbf{d}_1)$ are drawn i.i.d from $\mathsf{N}(\mathbf{0}, 2\mathbf{I}_5)$, $a_{j0}$ and elements of $\mathbf{a}_{j1}$ are drawn i.i.d from $\mathsf{N}(0, 2)$. For each source, we simulate 10 replications with $n_s = 1000$ records. We only keep $\{(y_i^s, w_i^s, \boldsymbol{x}_i^s)\}_{i=1}^{n_s}$ as the observed data, where $y_i^s = y_i^s(0)$ if $w_i^s = 0$ and $y_i^s = y_i^s(1)$ if $w_i^s = 1$. In each source, we use 50 data points for training, 450 for testing and 400 for validating. We report the evaluation metrics and their standard errors over the 10 replications.

**Result and discussion (I).** In the first experiment, we study the performance of CausalRFF on multiple sources whose data distributions are the same. To do that, we simulate $m = 5$ sources from the same distribution, i.e., we set the ground truth $\Delta = 0.0$ for all the sources. We refer to this dataset as DATA^same. In this experiment, we expect that the result of CausalRFF, which is trained in federated setting, is as good as training on combined data. The results in Figure 3 show that the error in two cases seem to move together in a correlated fashion, which verifies our hypothesis.

In addition, to study the performance of CausalRFF on the sources whose data distributions are different, we also simulate $m = 5$ sources. However, the first source is with $\Delta = 0.0$ and the other

Table 2: Out-of-sample errors on DATA$^{\text{diff}}$.

| Method | The error of CATE, $\sqrt{\epsilon_{\text{ATE}}}$ | | | The error of ATE, $\epsilon_{\text{ATE}}$ | | |
|---|---|---|---|---|---|---|
| | 1 source | 3 sources | 5 sources | 1 source | 3 sources | 5 sources |
| BART$_{\text{ag}}$ | - | **3.0**±**.01** | **3.0**±**.02** | - | 1.3±.05 | **1.4**±**.10** |
| X-Learner$_{\text{ag}}$ | - | 3.3±.03 | 3.3±.04 | - | **1.2**±**.09** | **1.3**±**.09** |
| R-Learner$_{\text{ag}}$ | - | 3.2±.03 | 3.1±.02 | - | **1.0**±**.07** | 1.2±.09 |
| OthoRF$_{\text{ag}}$ | - | 3.6±.05 | 3.6±.05 | - | 1.3±.09 | 1.6±.10 |
| TARNet$_{\text{ag}}$ | - | 6.1±.19 | 5.7±.05 | - | 2.5±.06 | 3.0±.05 |
| CFR-wass$_{\text{ag}}$ | - | 5.6±.09 | 5.7±.07 | - | 2.7±.05 | 2.8±.04 |
| CFR-mmd$_{\text{ag}}$ | - | 5.9±.08 | 5.6±.05 | - | 2.5±.03 | 2.8±.02 |
| CEVAE$_{\text{ag}}$ | - | 4.2±.07 | 3.9±.05 | - | 2.1±.09 | 1.8±.10 |
| BART$_{\text{cb}}$ | 3.1±.05 | 4.1±.10 | 4.2±.10 | 0.8±.17 | 2.8±.15 | 2.9±.14 |
| X-Learner$_{\text{cb}}$ | 3.3±.03 | 5.0±.08 | 4.6±.10 | **0.5**±**.12** | 3.3±.11 | 3.1±.13 |
| R-Learner$_{\text{cb}}$ | 3.3±.05 | 3.5±.05 | 3.3±.05 | 0.7±.18 | **1.1**±**.10** | **1.3**±**.10** |
| OthoRF$_{\text{cb}}$ | 3.9±.06 | 5.2±.10 | 4.6±.09 | **0.5**±**.11** | 3.3±.14 | 3.0±.12 |
| TARNet$_{\text{cb}}$ | 4.2±.07 | 5.9±.09 | 5.8±.06 | 2.2±.13 | 2.3±.04 | 2.9±.02 |
| CFR-wass$_{\text{cb}}$ | 4.0±.11 | 5.7±.08 | 5.5±.08 | 1.9±.06 | 2.4±.03 | 2.9±.04 |
| CFR-mmd$_{\text{cb}}$ | 3.8±.05 | 5.7±.08 | 5.5±.04 | 2.1±.04 | 2.4±.03 | 2.9±.04 |
| CEVAE$_{\text{cb}}$ | **2.4**±**.03** | 5.0±.06 | 4.4±.07 | **0.3**±**.08** | 2.6±.10 | 2.0±.07 |
| FedCI | **2.5**±**.03** | 2.6±.04 | 2.8±.04 | **0.2**±**.06** | 1.2±.12 | 1.5±.13 |
| CausalRFF | **1.4**±**.07** | **1.7**±**.12** | **1.9**±**.17** | **0.5**±**.11** | **1.1**±**.19** | **1.4**±**.27** |

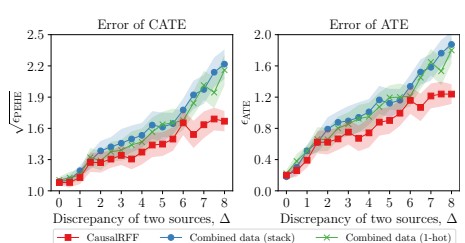

Figure 5: Experimental results on different levels of discrepancy, $\Delta$.

four sources are with $\Delta = 4.0$. We refer to this dataset as DATA$^{\text{diff}}$. We test the error of CATE and ATE on the first source. In this case, we expect that the errors of CausalRFF to be lower than that of training on combined data since CausalRFF learns the adaptive factors which prevent negative impact of the other four sources to the first source. The results in Figure 4 show that CausalRFF achieves lower errors compared to training on combined data (there are two cases of combining: stacking data, and adding one-hot vectors to indicate the source of each data point), which is as expected.

In the third experiment, we study the effect of $\Delta$ on the performance of CausalRFF. We simulate $m = 2$ sources with different values of $\Delta$. In particular, the first source is with $\Delta = 0.0$ and the second source is with $\Delta$ varying from 0.0 to 8.0. We compare our CausalRFF method with that of training on combined data. Again, Figure 5 shows that CausalRFF achieves lower errors as expected.

**Result and discussion (II).** This section aims to compare CausalRFF with the baselines on both datasets: DATA$^{\text{same}}$ and DATA$^{\text{diff}}$. Except FedCI (which is a Bayesian federated method), the other baselines are trained on two cases: combined data (cb) and bootstrap aggregating (ag) as mentioned earlier. On DATA$^{\text{same}}$, we expect that the performance of the proposed method is as good as the baselines trained on combined data. The results in Table 1 show that the performance of CausalRFF is as expected. For DATA$^{\text{diff}}$, we report the results on Table 2. The figures reveal that the performance of CausalRFF is as good as the baselines in predicting ATE. In terms of predicting CATE, the performance of the baselines significantly reduces as we add more data sources whose distribution are different from the first source. Meanwhile, the performance of CausalRFF in predicting CATE is slightly reduced, but it is still much better than those of the baselines. The reason of this is because we used adaptive factors to learn for the similarity of data distributions among the sources.

## 5.2 Large-scale Synthetic Data

**Data description.** In this section, we conduct experiments on a large number of sources. The set up in this section is similar to that of Section 5.1. We simulate two cases: (1) DATA-LARGE$^{\text{same}}$: a dataset of 100 sources, where we set $\Delta = 0$ for all sources so that their distributions are the same. (2) DATA-LARGE$^{\text{diff}}$: a dataset of 100 sources, where we draw uniformly the discrepancy factor $\Delta \sim \text{U}[0, 8]$ for each source so that their distributions are different. In both cases, we use test set from the first 20 sources for evaluation.

**Result and discussion.** Table 3 shows that CausalRFF achieves competitive results in estimating ATE and CATE when the sources have the same distribution. Table 4 shows that CausalRFF outperforms the baselines when the sources have different distributions. These results are consistent with our discussions in Section 5.1.

## 5.3 A Real World Dataset

**Data description.** The Infant Health and Development Program (IHDP) (Hill 2011) is a randomized study on the impact of specialist visits (the treatment) on the cognitive development of children (the

Table 3: Errors on DATA-LARGE$^{\text{same}}$ dataset.

| Method | The error of CATE, $\sqrt{\epsilon_{\text{ATE}}}$ | | | The error of ATE, $\epsilon_{\text{ATE}}$ | | |
|---|---|---|---|---|---|---|
| | 20 sources | 50 sources | 100 sources | 20 sources | 50 sources | 100 sources |
| BART$_{\text{cb}}$ | 3.4±.03 | 3.4±.01 | 3.3±.01 | 1.4±.06 | 1.3±.02 | 1.3±.01 |
| X-Learner$_{\text{cb}}$ | 3.0±.01 | 2.9±.01 | 2.9±.01 | **.16±.02** | **.12±.02** | **.13±.02** |
| R-Learner$_{\text{cb}}$ | 3.0±.01 | 2.9±.01 | 2.9±.01 | **.07±.01** | **.10±.02** | **.10±.02** |
| OthoRF$_{\text{cb}}$ | 3.4±.03 | 3.3±.01 | 3.2±.01 | 1.2±.06 | 1.1±.02 | 1.0±.02 |
| TARNet$_{\text{cb}}$ | 3.8±.03 | 3.7±.01 | 3.3±.01 | 1.1±.02 | 1.0±.01 | .93±.01 |
| CFR-wass$_{\text{cb}}$ | 3.7±.02 | 3.6±.01 | 3.2±.01 | 1.1±.02 | .99±.01 | .87±.01 |
| CFR-mmd$_{\text{cb}}$ | 3.7±.02 | 3.6±.01 | 3.2±.01 | 1.1±.02 | .98±.01 | .87±.01 |
| CEVAE$_{\text{cb}}$ | **2.3±.01** | **2.2±.01** | **2.0±.01** | .19±.03 | .17±.01 | .17±.01 |
| FedCI | **2.2±.02** | **2.2±.01** | **1.9±.01** | .23±.04 | .21±.01 | .19±.01 |
| CausalRFF | **1.6±.05** | **1.6±.01** | **1.5±.01** | 0.3±.04 | 0.2±.02 | **.16±.02** |

Table 4: Errors on DATA-LARGE$^{\text{diff}}$ dataset.

| Method | The error of CATE, $\sqrt{\epsilon_{\text{PEHE}}}$ | | | The error of ATE, $\epsilon_{\text{ATE}}$ | | |
|---|---|---|---|---|---|---|
| | 20 sources | 50 sources | 100 sources | 20 sources | 50 sources | 100 sources |
| BART$_{\text{cb}}$ | 3.4±.03 | 3.5±.01 | 3.5±.01 | 1.4±.06 | 1.5±.02 | 1.5±.01 |
| X-Learner$_{\text{cb}}$ | 3.3±.04 | **3.2±.01** | 3.2±.01 | 1.1±.08 | 1.2±.02 | 1.2±.02 |
| R-Learner$_{\text{cb}}$ | 3.2±.03 | 3.1±.01 | 3.1±.01 | **.88±.07** | **.88±.02** | **.86±.01** |
| OthoRF$_{\text{cb}}$ | 3.4±.03 | 3.4±.01 | 3.4±.01 | 1.2±.07 | **1.2±.02** | 1.3±.01 |
| TARNet$_{\text{cb}}$ | 5.6±.04 | 5.6±.02 | 5.7±.02 | 2.7±.06 | 2.8±.02 | 2.8±.02 |
| CFR-wass$_{\text{cb}}$ | 5.4±.05 | 5.5±.02 | 5.5±.02 | 2.7±.05 | 2.7±.02 | 2.7±.02 |
| CFR-mmd$_{\text{cb}}$ | 5.4±.05 | 5.4±.02 | 5.5±.02 | 2.7±.05 | 2.7±.02 | 2.7±.02 |
| CEVAE$_{\text{cb}}$ | 3.4±.04 | 3.4±.02 | 3.3±.01 | 1.2±.06 | **1.2±.02** | **1.2±.01** |
| FedCI | **3.2±.03** | 3.2±.02 | 3.0±.01 | 1.2±.07 | **1.2±.01** | **1.2±.01** |
| CausalRFF | **1.8±.03** | **1.7±.03** | **1.6±.01** | .24±.04 | .19±.14 | .15±.01 |

outcome). The dataset consists of 747 records with 25 covariates describing properties of the children and their mothers. The treatment group includes children who received specialist visits and control group includes children who did not receive. This dataset was 'de-randomized' by removing from the treated set children with non-white mothers. For each child, a treated and a control outcome are then simulated, thus allowing us to know the 'true' individual causal effects of the treatment. Further details are presented in Appendix.

**Result and discussion.** Table 5 reports the experimental results on IHDP dataset. Again, we see that the proposed method gives competitive results compared to the baselines. In particular, the error of CausalRFF in predicting ATE is as low as that of the baselines, which is as we expected. In addition, the errors of CausalRFF in predicting CATE are lower than those of the baselines, which verifies the efficacy of the proposed method. Most importantly, CausalRFF is trained in a federated setting which minimizes the risk of privacy breach for the individuals stored in the local dataset.

Table 5: Out-of-sample errors on IHDP dataset.

| Method | The error of CATE, $\sqrt{\epsilon_{\text{PEHE}}}$ | | | The error of ATE, $\epsilon_{\text{ATE}}$ | | |
|---|---|---|---|---|---|---|
| | 1 source | 2 sources | 3 sources | 1 source | 2 sources | 3 sources |
| BART$_{\text{ag}}$ | - | 2.3±.26 | 2.4±.22 | - | 1.2±.23 | 1.3±.18 |
| X-Learner$_{\text{ag}}$ | - | **1.8±.20** | 1.8±.22 | - | **0.6±.15** | **0.4±.11** |
| R-Learner$_{\text{ag}}$ | - | 2.4±.31 | 2.3±.21 | - | 1.3±.34 | 1.2±.24 |
| OthoRF$_{\text{ag}}$ | - | 2.3±.21 | 2.1±.16 | - | 0.6±.22 | 0.7±.13 |
| TARNet$_{\text{ag}}$ | - | 2.9±.13 | 2.7±.15 | - | **0.7±.12** | 0.7±.16 |
| CFR-wass$_{\text{ag}}$ | - | 2.3±.31 | 2.2±.20 | - | **0.7±.12** | 0.7±.11 |
| CFR-mmd$_{\text{ag}}$ | - | 2.6±.21 | 2.4±.15 | - | 0.8±.19 | 0.7±.18 |
| CEVAE$_{\text{ag}}$ | - | 1.9±.14 | **1.6±.17** | - | 1.2±.11 | 0.8±.10 |
| BART$_{\text{cb}}$ | 2.2±.22 | 2.1±.26 | 2.1±.25 | 1.0±.16 | 0.8±.20 | 0.7±.17 |
| X-Learner$_{\text{cb}}$ | 1.9±.21 | 1.9±.21 | 1.8±.18 | **0.5±.21** | **0.5±.18** | **0.4±.11** |
| R-Learner$_{\text{cb}}$ | 2.8±.31 | 2.6±.23 | 2.6±.17 | 1.6±.25 | 1.6±.26 | 1.6±.19 |
| OthoRF$_{\text{cb}}$ | 2.8±.16 | 2.1±.14 | 1.9±.14 | **0.8±.15** | **0.6±.10** | **0.6±.10** |
| TARNet$_{\text{cb}}$ | 3.5±.59 | 2.7±.12 | 2.5±.15 | 1.6±.61 | **0.7±.12** | **0.6±.17** |
| CFR-wass$_{\text{cb}}$ | 2.2±.15 | 2.1±.22 | 2.1±.23 | **0.7±.23** | **0.6±.18** | **0.6±.16** |
| CFR-mmd$_{\text{cb}}$ | 2.7±.19 | 2.3±.26 | 2.2±.10 | 0.9±.30 | **0.7±.17** | **0.5±.17** |
| CEVAE$_{\text{cb}}$ | **1.8±.22** | 2.0±.11 | **1.7±.12** | **0.5±.14** | 1.4±.07 | 0.9±.07 |
| FedCI | **1.6±.10** | **1.6±.12** | 1.7±.09 | **0.5±.10** | **0.5±.24** | **0.5±.09** |
| CausalRFF | 1.7±.34 | **1.4±.33** | **1.2±.18** | 0.7±.14 | 0.7±.17 | **0.5±.16** |

## 6 Conclusion

We have proposed a new method to learn causal effects from federated, observational data sources with dissimilar distributions. Our method utilizes Random Fourier Features that naturally induce the decomposition of the loss function to individual components. Our method allows for each component data group to inherit different distributions, and requires no prior knowledge on data discrepancy among the sources. We have also proved statistical guarantees which show how multiple data sources are effectively incorporated in our causal model. Our work is an important step toward privacy-preserving causal inference. Future work may include combining the proposed method with a multiparty differential privacy technique (e.g., Pathak et al. 2010; Rajkumar and Agarwal 2012; Pettai and Laud 2015; Hamm et al. 2016), which might lead to a stronger privacy guarantee model. Another direction is to extend the proposed method with some recent ideas (e.g., Khemakhem et al. 2020; Sun et al. 2021) to study the identifiability of the model.

## Acknowledgments and Disclosure of Funding

This research/project is supported by the National Research Foundation Singapore and DSO National Laboratories under the AI Singapore Programme (AISG Award No: AISG2-RP-2020-016).

AB was partially supported by an NRF Fellowship for AI (NRF-NRFAI11-2019-0002) and an Amazon Research Award.

This work was conducted while YL was at Harvard University and the views expressed here do not necessarily reflect the position of Roche AG.

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
