# Appendix:
# An Adaptive Kernel Approach to Federated Learning of Heterogeneous Causal Effects

**Thanh Vinh Vo**[1]     **Arnab Bhattacharyya**[1]     **Young Lee**[2]     **Tze-Yun Leong**[1]

[1]School of Computing, National University of Singapore
[2]Roche AG and Harvard University
{votv,arnabb,leongty}@nus.edu.sg

## Contents

36th Conference on Neural Information Processing Systems (NeurIPS 2022).

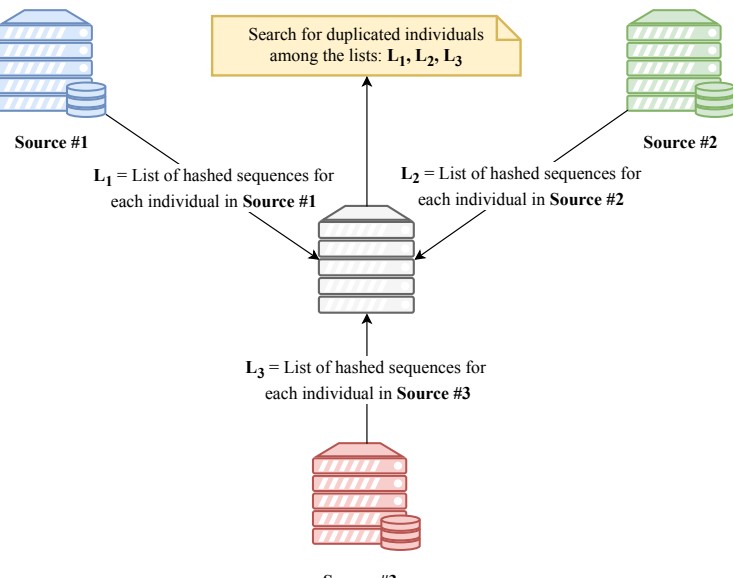

Figure 1: An illustration on how the pre-training step. This step is intended to identify duplicated individuals among the sources. Furthermore, this step preserves privacy since each source sends only their hashed sequences of the individuals.

## A  Pre-training step to remove duplicated individuals

As mentioned in the main text, we make five assumptions as follows:

**(A1)** *Consistency*: $W = w \implies Y(w) = Y$, this follows from the axioms of structural causal model.

**(A2)** *No interference*: treatment on one subject does not affect the outcomes of another one. This is because the outcome only has a single node for treatment as a parent.

**(A3)** *Positivity* (also known as *Overlap*): every subject has some positive probability to be assigned to every treatment.

**(A4)** The individuals in each source must have the same set of *common* covariates.

**(A5)** There is no individual whose data exists in more than one source.

Assumptions **(A1)**, **(A2)** and **(A3)** are standard in any causal inference algorithm.

Assumption **(A4)** has been implicitly shown in our setup since all the sources would share the same causal graph. This is a reasonable assumption as we intend to build a unified model on all of the data sources. For example, decentralized data in Choudhury et al. (2019); Vaid et al. (2020); Flores et al. (2020) (to name a few) satisfy this assumption for federated learning.

Assumption **(A5)** is to ensure that no individuals would dominate the other individuals when training the model. For example, if an individual appears in all of the sources, the trained model would be biased by data of this individual (there is imbalance caused by the use of more data from this particular individual than the others). Hence, this condition would ensure that such bias does not exist.

In practice, Assumption **(A5)** sometimes does not hold. To address such a problem, we propose a pre-training step to exclude such duplicated individuals. The pre-training step are summarized as follows:

**(1)** Suppose that an individual can be uniquely identified via a set of features. For example, a pair of (national identity, nationality) can be used to uniquely identify a person.

**(2)** To identify duplicated individuals, we first encode the above features with a hash function such as MD5, SHA256.

**(3)** We then send the encoded sequences to a central server.

**(4)** The server would collect all encoded sequences from all sources and find among them if an encoded sequence is repeated.

**(5)** All of the repeated sequences are associated with duplicated individuals. Thus, we announce the sources to exclude these individual from the training process.

We summarize the pre-training step in Figure 1 with three sources of data.

## B  Identification

The causal effects are unidentifiable if the confounders are unobserved. However, Louizos et al. (2017) showed that if the joint distribution $p_\mathsf{s}(\boldsymbol{x}^\mathsf{s}, y^\mathsf{s}, w^\mathsf{s}, \boldsymbol{z}^\mathsf{s})$ can be recovered, then the causal effects are identifiable. In the following, we show how they are identifiable.

*Proof.* The proof is adapted from Louizos et al. (Theorem 1, 2017). We need to show that the distribution $p_\mathsf{s}(y^\mathsf{s}|\mathrm{do}(W = \boldsymbol{w}^\mathsf{s}), \boldsymbol{x}^\mathsf{s})$ is identifiable from observational data. We have

$$p_\mathsf{s}(y^\mathsf{s}|\mathrm{do}(W = \boldsymbol{w}^\mathsf{s}), \boldsymbol{x}^\mathsf{s}) = \int p_\mathsf{s}(y^\mathsf{s}|\mathrm{do}(W = \boldsymbol{w}^\mathsf{s}), \boldsymbol{x}^\mathsf{s}, \boldsymbol{z}^\mathsf{s}) p_\mathsf{s}(\boldsymbol{z}^\mathsf{s}|\mathrm{do}(W = \boldsymbol{w}^\mathsf{s}), \boldsymbol{x}^\mathsf{s}) d\boldsymbol{z}^\mathsf{s}$$

$$= \int p_\mathsf{s}(y^\mathsf{s}|\boldsymbol{w}^\mathsf{s}, \boldsymbol{x}^\mathsf{s}, \boldsymbol{z}^\mathsf{s}) p_\mathsf{s}(\boldsymbol{z}^\mathsf{s}|\boldsymbol{x}^\mathsf{s}) d\boldsymbol{z}^\mathsf{s}.$$

where the last equality is obtained by applying the *do*-calculus. The last expression, $\int p_\mathsf{s}(y^\mathsf{s}|\boldsymbol{w}^\mathsf{s}, \boldsymbol{x}^\mathsf{s}, \boldsymbol{z}^\mathsf{s}) p_\mathsf{s}(\boldsymbol{z}^\mathsf{s}|\boldsymbol{x}^\mathsf{s}) d\boldsymbol{z}^\mathsf{s}$, can be identified by the joint distribution $p_\mathsf{s}(\boldsymbol{x}^\mathsf{s}, y^\mathsf{s}, w^\mathsf{s}, \boldsymbol{z}^\mathsf{s})$. In our work, $p_\mathsf{s}(\boldsymbol{x}^\mathsf{s}, y^\mathsf{s}, w^\mathsf{s}, \boldsymbol{z}^\mathsf{s})$ is recovered by its factorization with the distributions $p_\mathsf{s}(w^\mathsf{s}|\boldsymbol{x}^\mathsf{s})$, $p_\mathsf{s}(y^\mathsf{s}|\boldsymbol{x}^\mathsf{s}, w^\mathsf{s})$, $p_\mathsf{s}(\boldsymbol{z}^\mathsf{s}|\boldsymbol{x}^\mathsf{s}, y^\mathsf{s}, w^\mathsf{s})$, $p_\mathsf{s}(y^\mathsf{s}|w^\mathsf{s}, \boldsymbol{z}^\mathsf{s})$, and $p(\boldsymbol{z}^\mathsf{s})$. Adaptively learning these distributions in a federated setting is the main task of our work. This completes the proof. □

## C  Computing CATE, local ATE, and global ATE

This section gives details on how to compute CATE, local ATE and global ATE after training the model.

### C.1  Computing the CATE and local ATE

After training the model, each source *can* compute the CATE and the local ATE on for its own source and use it for itself.

$$E[y_i^\mathsf{s}|\mathrm{do}(w_i^\mathsf{s}{=}w), \boldsymbol{x}_i^\mathsf{s}] = \int E[y_i^\mathsf{s}|w_i^\mathsf{s}{=}w, \boldsymbol{z}_i^\mathsf{s}] p(\boldsymbol{z}_i^\mathsf{s}|\boldsymbol{x}_i^\mathsf{s}) d\boldsymbol{z}_i^\mathsf{s} \simeq \frac{1}{N} \sum_{l=1}^{N} f_y(w_i^\mathsf{s}{=}w, \boldsymbol{z}_i^\mathsf{s}[l])$$

where $f_y(w_i^\mathsf{s}{=}w, \boldsymbol{z}_i^\mathsf{s}[l])$ is the mean function of $p_\mathsf{s}(y_i^\mathsf{s}|w_i^\mathsf{s}, \boldsymbol{z}_i^\mathsf{s})$ and $\{\boldsymbol{z}_i^\mathsf{s}[l]\}_{l=1}^{N} \overset{i.i.d.}{\sim} p_\mathsf{s}(\boldsymbol{z}_i^\mathsf{s}|\boldsymbol{x}_i^\mathsf{s})$.

The problem is to draw $\{\boldsymbol{z}_i^\mathsf{s}[l]\}_{l=1}^{N}$ from $p_\mathsf{s}(\boldsymbol{z}_i^\mathsf{s}|\boldsymbol{x}_i^\mathsf{s})$. We observe that

$$p_\mathsf{s}(\boldsymbol{z}_i^\mathsf{s}|\boldsymbol{x}_i^\mathsf{s}) = \sum_{w_i^\mathsf{s} \in \{0,1\}} \int p_\mathsf{s}(\boldsymbol{z}_i^\mathsf{s}|\boldsymbol{x}_i^\mathsf{s}, y_i^\mathsf{s}, w_i^\mathsf{s}) p_\mathsf{s}(y_i^\mathsf{s}|\boldsymbol{x}_i^\mathsf{s}, w_i^\mathsf{s}) p_\mathsf{s}(w_i^\mathsf{s}|\boldsymbol{x}_i^\mathsf{s}) \, dy_i^\mathsf{s}.$$

Hence, to draw samples, we proceed in the following steps:

**(1)** Draw a sample of $w_i^\mathsf{s}$ from $p_\mathsf{s}(w_i^\mathsf{s}|\boldsymbol{x}_i^\mathsf{s})$.

**(2)** Substitute the above sample of $w_i^\mathsf{s}$ to $p_\mathsf{s}(y_i^\mathsf{s}|\boldsymbol{x}_i^\mathsf{s}, w_i^\mathsf{s})$.

**(3)** Draw a sample of $y_i^\mathsf{s}$ from $p_\mathsf{s}(y_i^\mathsf{s}|\boldsymbol{x}_i^\mathsf{s}, w_i^\mathsf{s})$.

**(4)** Substitute the above sample of $y_i^\mathsf{s}$ to $p_\mathsf{s}(\boldsymbol{z}_i^\mathsf{s}|\boldsymbol{x}_i^\mathsf{s}, y_i^\mathsf{s}, w_i^\mathsf{s})$.

**(5)** Draw a sample of $\boldsymbol{z}_i^\mathsf{s}$ from $p_\mathsf{s}(\boldsymbol{z}_i^\mathsf{s}|\boldsymbol{x}_i^\mathsf{s}, y_i^\mathsf{s}, w_i^\mathsf{s})$.

The density function of $p_{\mathsf{s}}(y_i^{\mathsf{s}}|\boldsymbol{x}_i^{\mathsf{s}}, w_i^{\mathsf{s}})$ and $p_{\mathsf{s}}(w_i^{\mathsf{s}}|\boldsymbol{x}_i^{\mathsf{s}})$ are available after training the model. As described in the main text, there are two options to draw from $p_{\mathsf{s}}(\boldsymbol{z}_i^{\mathsf{s}}|\boldsymbol{x}_i^{\mathsf{s}}, y_i^{\mathsf{s}}, w_i^{\mathsf{s}})$. The first option is to draw from $q(\boldsymbol{x}_i^{\mathsf{s}})$ sine it approximates $p_{\mathsf{s}}(\boldsymbol{z}_i^{\mathsf{s}}|\boldsymbol{x}_i^{\mathsf{s}}, y_i^{\mathsf{s}}, w_i^{\mathsf{s}})$. The second option is to use Metropolis-Hastings algorithm with independent sampler (Liu 1996). For the second option, we have that

$$p_{\mathsf{s}}(\boldsymbol{z}_i^{\mathsf{s}}|\boldsymbol{x}_i^{\mathsf{s}}, y_i^{\mathsf{s}}, w_i^{\mathsf{s}}) \propto p_{\mathsf{s}}(y_i^{\mathsf{s}}|\boldsymbol{z}_i^{\mathsf{s}}, w_i^{\mathsf{s}}) p_{\mathsf{s}}(w_i^{\mathsf{s}}|\boldsymbol{z}_i^{\mathsf{s}}) p_{\mathsf{s}}(\boldsymbol{x}_i^{\mathsf{s}}|\boldsymbol{z}_i^{\mathsf{s}}) p(\boldsymbol{z}_i^{\mathsf{s}}).$$

Hence, it can be used to compute the acceptance probability of interest. Note that the second option would give more exact samples since it further filters the samples based on the exact acceptance probability.

The above would help estimate the CATE given $\boldsymbol{x}_i^{\mathsf{s}}$. The local ATE is the average of CATE of individuals in a source s. These quantities can be estimated in a local source's machine. We show how to compute the global ATE in the next section.

### C.2 Computing the global ATE from local ATE of each Source

To compute a global ATE, the server would collect all the local ATE in each source and then compute their weighted average. For example, suppose that we have three sources whose local ATE values are 7.0, 8.5, and 6.8. These local ATEs are averaged over 10, 5, and 12 individuals, in that order. Then, the global ATE is given as follows:

$$\text{global ATE} = \frac{10 \times 7.0 + 8 \times 8.5 + 12 \times 6.8}{10 + 8 + 12} = 7.32.$$

Since each source only shares their local ATE and the number of individuals, it does not leak any sensitive information about the individuals.

## D Comparison metrics

We report two error metrics in our experiments:

- Precision in estimation of heterogeneous effects (PEHE):

$$\epsilon_{\text{PEHE}} = \sum_{i=1}^{n} (\tau(\boldsymbol{x}_i) - \hat{\tau}(\boldsymbol{x}_i))^2 / n, \tag{1}$$

- Absolute error:

$$\epsilon_{\text{ATE}} = |\tau - \hat{\tau}|, \tag{2}$$

where $\tau(\boldsymbol{x}_i), \tau$ are the ground truth of ITE and ATE, and $\hat{\tau}(\boldsymbol{x}_i), \hat{\tau}$ are their estimates. We report the mean and standard error over 10 replicates of the data with different random initializations of the training algorithm.

## E Derivation of the loss functions

In this section, we present the loss functions and the form of functions that modulate the desired distributions.

### E.1 Learning distributions involving latent confounder

The ELBO of the log marginal likelihood has the following expression

$$\log p(\mathbf{x}, \mathbf{y}, \mathbf{w}) = \log \int p(\mathbf{x}, \mathbf{y}, \mathbf{w}, \mathbf{z}) d\mathbf{z}$$

$$\geq \int q(\mathbf{z}) \log \frac{p(\mathbf{x}, \mathbf{y}, \mathbf{w}, \mathbf{z})}{q(\mathbf{z})} d\mathbf{z}$$

$$= \sum_{\mathsf{s} \in \mathcal{S}} \sum_{i=1}^{n_{\mathsf{s}}} \left( E_q \big[ \log p_{\mathsf{s}}(y_i^{\mathsf{s}}|w_i^{\mathsf{s}}, \boldsymbol{z}_i^{\mathsf{s}}) + \log p_{\mathsf{s}}(w_i^{\mathsf{s}}|\boldsymbol{z}_i^{\mathsf{s}}) + \log p_{\mathsf{s}}(\boldsymbol{x}_i^{\mathsf{s}}|\boldsymbol{z}_i^{\mathsf{s}}) \big] - \text{KL}[q(\boldsymbol{z}_i^{\mathsf{s}}) \| p(\boldsymbol{z}_i^{\mathsf{s}})] \right) =: \mathcal{L}.$$

Using the complete dataset $\tilde{D}^s = \bigcup_{l=1}^{M} \left\{ (w_i^s, y_i^s, \boldsymbol{x}_i^s, \boldsymbol{z}_i^s[l]) \right\}_{i=1}^{n_s}, \forall s \in \mathcal{S}$, we minimize the following loss function $J$:

$$J = \widehat{\mathcal{L}} + \sum_{c \in \mathcal{A}} R(f_c), \qquad \mathcal{A} = \{y_0, y_1, q_0, q_q, x, w\},$$

where $\widehat{\mathcal{L}}$ is the empirical loss function obtained from the negative of $\mathcal{L}$. In the following, we find the form of $f_c$ based on the representer theorem.

We further define $f_x = [f_{x,1}, ..., f_{x,d_x}]$, where $f_{x,d}$ is a function taking $\boldsymbol{z}_i^s$ as input and mapping it to a real value in $\mathbb{R}$. Similarly, $f_{q0} = [f_{q0,1}, ..., f_{q0,d_z}]$ and $f_{q1} = [f_{q1,1}, ..., f_{q1,d_z}]$.

Let $\mathcal{H}_c$ ($c \in \mathcal{A}$) be a reproducing Kernel Hilbert space (RKHS) and $\kappa_c(\cdot, \cdot)$ be kernel function associated with $\mathcal{H}_c$. We define $\mathcal{B}_c$ as follows:

$$\mathcal{B}_{y_0} = \mathrm{span}\big\{\kappa_{y0}(\cdot, \boldsymbol{z}_i^s[l]), \text{ where } s \in \mathcal{S}; i = 1, ..., n_s; l = 1, ..., M\big\},$$
$$\mathcal{B}_{y_1} = \mathrm{span}\big\{\kappa_{y1}(\cdot, \boldsymbol{z}_i^s[l]), \text{ where } s \in \mathcal{S}; i = 1, ..., n_s; l = 1, ..., M\big\},$$
$$\mathcal{B}_x = \mathrm{span}\big\{\kappa_x(\cdot, \boldsymbol{z}_i^s[l]), \text{ where } s \in \mathcal{S}; i = 1, ..., n_s; l = 1, ..., M\big\},$$
$$\mathcal{B}_w = \mathrm{span}\big\{\kappa_w(\cdot, \boldsymbol{z}_i^s[l]), \text{ where } s \in \mathcal{S}; i = 1, ..., n_s; l = 1, ..., M\big\},$$
$$\mathcal{B}_{q_0} = \mathrm{span}\big\{\kappa_{q0}(\cdot, [\boldsymbol{x}_i^s, y_i^s]), \text{ where } s \in \mathcal{S}; i = 1, ..., n_s\big\},$$
$$\mathcal{B}_{q_1} = \mathrm{span}\big\{\kappa_{q1}(\cdot, [\boldsymbol{x}_i^s, y_i^s]), \text{ where } s \in \mathcal{S}; i = 1, ..., n_s\big\}.$$

We posit the following regularizers:

$$R(f_{y0}) = \mathsf{reg\_factor}_{y0} \times \|f_{y0}\|_{\mathcal{H}_{y0}}^2, \quad R(f_x) = \sum_{d=1}^{d_x} \mathsf{reg\_factor}_{x,d} \times \|f_{x,d}\|_{\mathcal{H}_x}^2 \quad (d = 1, ..., d_x).$$

The regularizers $R(f_{y1})$ and $R(f_w)$ are similar to that of $R(f_{y0})$, and $R(f_{q0})$, $R(f_{q1})$ are similar to that of $R(f_x)$.

We see that $\mathcal{B}_c$ is a subspace of $\mathcal{H}_c$. We project $f_{y0}, f_{y1}, f_w, f_{x,d}$ ($d = 1, ..., d_x$), $f_{q0,d}$ ($d = 1, ..., d_z$) and $f_{q1,d}$ ($d = 1, ..., d_z$) onto the subspaces $\mathcal{B}_{y0}, \mathcal{B}_{y1}, \mathcal{B}_w, \mathcal{B}_x, \mathcal{B}_{q0}$ and $\mathcal{B}_{q1}$, respectively, and obtain $f_{y0}', f_{y1}', f_w', f_{x,d}', f_{q0,d}'$ and $f_{q1,d}'$. Next, we also project them onto the perpendicular spaces of $\mathcal{B}_{(\cdot)}$ to obtain $f_{y0}^\perp, f_{y1}^\perp, f_w^\perp, f_{x,d}^\perp, f_{q0,d}^\perp$ and $f_{q1,d}^\perp$.

Note that $f_{(\cdot)} = f_{(\cdot)}' + f_{(\cdot)}^\perp$. Hence, $\|f_{(\cdot)}\|_{\mathcal{H}_{(\cdot)}}^2 = \|f_{(\cdot)}'\|_{\mathcal{H}_{(\cdot)}}^2 + \|f_{(\cdot)}^\perp\|_{\mathcal{H}_{(\cdot)}}^2 \geq \|f_{(\cdot)}'\|_{\mathcal{H}_{(\cdot)}}^2$, which implies that $\mathsf{reg\_factor}_{(\cdot)} \times \|f_{(\cdot)}\|_{\mathcal{H}_{(\cdot)}}^2$ is minimized if $f_{(\cdot)}$ is in its subspace $\mathcal{B}_{(\cdot)}$. **(I)**

In addition, due to the reproducing property, we have

$$f_{y_0}(\boldsymbol{z}_i^s[l]) = \big\langle f_{y_0}, \kappa_{y_0}(\cdot, \boldsymbol{z}_i^s[l]) \big\rangle_{\mathcal{H}_y} = \big\langle f_{y_0}', \kappa_{y_0}(\cdot, \boldsymbol{z}_i^s[l]) \big\rangle_{\mathcal{H}_y} + \big\langle f_{y_0}^\perp, \kappa_{y_0}(\cdot, \boldsymbol{z}_i^s[l]) \big\rangle_{\mathcal{H}_y} = f_{y_0}'(\boldsymbol{z}_i^s[l]).$$

Similarly, we also have $f_{y1}(\boldsymbol{z}_i^d[l]) = f_{y1}'(\boldsymbol{z}_i^d[l])$, $f_w(\boldsymbol{z}_i^d[l]) = f_w'(\boldsymbol{z}_i^d[l])$, $f_{x,d}(\boldsymbol{z}_i^l) = f_{x,d}'(\boldsymbol{z}_i^d[l])$, $f_{q0,d}(y_i^d, \boldsymbol{x}_i^d) = f_{q0,d}'(y_i^d, \boldsymbol{x}_i^d)$ and $f_{q1,d}(y_i^d, \boldsymbol{x}_i^d) = f_{q1,d}'(y_i^d, \boldsymbol{x}_i^d)$. Hence,

$$\widehat{\mathcal{L}}(f_{y0}, f_{y1}, f_{q0}, f_{q1}, f_x, f_w) = \widehat{\mathcal{L}}(f_{y0}', f_{y1}', f_{q0}', f_{q1}', f_x', f_w'). \tag{**(II)**}$$

**(I)** and **(II)** imply that $f_{y0}, f_{y1}, f_{q0,d}, f_{q1,d}, f_{x,d}, f_w$ are the weighted sum of elements in their corresponding subspace. Hence,

$$\boxed{f_c(\boldsymbol{u}^s) = \sum_{v \in \mathcal{S}} \sum_{j=1}^{n_v \times M} \kappa(\boldsymbol{u}^s, \boldsymbol{u}_j^v) \boldsymbol{\alpha}_j^v.}$$

Using this form with the adaptive kernel and Random Fourier Feature described in the main text (Section 4.1), we obtain the desired model.

## E.2 Learning auxiliary distributions

The derivation of $J_w$, $J_y$ and the form of functions modulated the auxiliary distributions are similar to those of $J$ as detailed in Section E.1. The difference is that the empirical loss functions are obtained from the negative log-likelihood instead of the ELBO.

# F  Spectral distribution of some popular kernels

Table 1 (adopted from Milton et al. (2019)) presents some popular kernels and their associated spectral density $s(\boldsymbol{\omega})$. Those density functions are needed to draw samples of $\boldsymbol{\omega}$ for Random Fourier Features presented in Section 4 of the main text. In our experiments, we used Gaussian kernel.

Table 1: Some popular kernels and their associated spectral density. Note that $K_\nu(\cdot)$ denotes the modified Bessel function of the second kind, $\Gamma(\cdot)$ is the gamma function.

| Kernel | Kernel function, $k(\boldsymbol{x}_1 - \boldsymbol{x}_2)$ | Spectral density, $s(\boldsymbol{\omega})$ |
|---|---|---|
| Gaussian | $\exp\left(-\dfrac{\|\boldsymbol{x}_1 - \boldsymbol{x}_2\|_2^2}{2\ell^2}\right)$ | $\left(\dfrac{2\pi}{\ell^2}\right)^{\frac{-d}{2}} \exp\left(-\dfrac{\ell^2\|\boldsymbol{\omega}\|_2^2}{2}\right)$ |
| Laplacian | $\exp\left(-\ell\|\boldsymbol{x}_1 - \boldsymbol{x}_2\|_1\right)$ | $\left(\dfrac{2}{\pi}\right)^{\frac{d}{2}} \displaystyle\prod_{i=1}^{d} \dfrac{\ell}{\ell^2 + \omega_i^2}$ |
| Matérn | $\dfrac{2^{1-\nu}}{\Gamma(\nu)}\left(\sqrt{2\nu}\dfrac{\|\boldsymbol{x}_1 - \boldsymbol{x}_2\|_2}{\ell}\right)^\nu K_\nu\left(\sqrt{2\nu}\dfrac{\|\boldsymbol{x}_1 - \boldsymbol{x}_2\|_2}{\ell}\right)$ | $\dfrac{2^d\pi^{\frac{d}{2}}\Gamma(\nu+\frac{d}{2})(2\nu)^\nu}{\Gamma(\nu)\ell^{2\nu}}\left(\dfrac{2\nu}{\ell^2} + 4\pi^2\|\boldsymbol{\omega}\|_2^2\right)^{-\left(\nu+\frac{d}{2}\right)}$ |

# G  Proof of Lemma 1

We repeat Lemma 1 here for convenience:

**Lemma 1** (With presence of latent variables). *Let* $\boldsymbol{\theta} = \{\theta^{\mathsf{s}}\}_{\mathsf{s}=1}^m$ *and* $\hat{\boldsymbol{\theta}}$ *be its estimate. Let* $y_i^{\mathsf{s}} \in \mathbb{R}$ *and* $\boldsymbol{x}_i^{\mathsf{s}} \in \mathbb{R}^{d_x}$. *Then,*

$$\inf_{\hat{\boldsymbol{\theta}}} \sup_{P \in \mathcal{P}} \mathbb{E}_P\left[\|\hat{\boldsymbol{\theta}} - \boldsymbol{\theta}(P)\|_2\right] \geq \frac{\sqrt{m(d_x+3)}\log(2\sqrt{m})}{64\sqrt{B}\sum_{\mathsf{s}\in\mathcal{S}} n_{\mathsf{s}}\left(1 + \sum_{\mathsf{v}\in\mathcal{S}, \mathsf{v}\neq\mathsf{s}} \lambda^{\mathsf{s},\mathsf{v}}\right)^2}.$$

Let $\mathcal{S}_{\backslash \mathsf{s}} := \mathcal{S} \setminus \{\mathsf{s}\}$. The model is summarized as follows:

$$p(\boldsymbol{z}_i^{\mathsf{s}}) = \mathsf{N}(0, \sigma_z^2 \mathbf{I}_{d_z}),$$

$$p(w_i^{\mathsf{s}}|\boldsymbol{z}_i^{\mathsf{s}}) = \mathsf{Bern}\left(\varphi\left(\left(\theta_w^{\mathsf{s}} + \sum_{\mathsf{v}\in\mathcal{S}_{\backslash\mathsf{s}}} \lambda^{\mathsf{s},\mathsf{v}}\theta_w^{\mathsf{v}}\right)^\top \phi(\boldsymbol{z}_i^{\mathsf{s}})\right)\right),$$

$$p(y_i^{\mathsf{s}}|w_i^{\mathsf{s}},\boldsymbol{z}_i^{\mathsf{s}}) = \mathsf{N}\left(\left(w_i^{\mathsf{s}}\left(\theta_{y1}^{\mathsf{s}} + \sum_{\mathsf{v}\in\mathcal{S}_{\backslash\mathsf{s}}} \lambda^{\mathsf{s},\mathsf{v}}\theta_{y1}^{\mathsf{v}}\right) + (1-w_i^{\mathsf{s}})\left(\theta_{y0}^{\mathsf{s}} + \sum_{\mathsf{v}\in\mathcal{S}_{\backslash\mathsf{s}}} \lambda^{\mathsf{s},\mathsf{v}}\theta_{y0}^{\mathsf{v}}\right)\right)^\top \phi(\boldsymbol{z}_i^{\mathsf{s}}), \sigma_y^2\right),$$

$$p(\boldsymbol{x}_i^{\mathsf{s}}|\boldsymbol{z}_i^{\mathsf{s}}) = \mathsf{N}\left(\left(\theta_x^{\mathsf{s}} + \sum_{\mathsf{v}\in\mathcal{S}_{\backslash\mathsf{s}}} \lambda^{\mathsf{s},\mathsf{v}}\theta_x^{\mathsf{v}}\right)^\top \phi(\boldsymbol{z}_i^{\mathsf{s}}), \sigma_x^2\mathbf{I}_{d_x}\right),$$

where $\boldsymbol{z}_i^{(\cdot)} \in \mathbb{R}^{d_z}$, $y_i^{(\cdot)} \in \mathbb{R}$, $w_i^{(\cdot)} \in \{0,1\}$, $\boldsymbol{x}_i^{(\cdot)} \in \mathbb{R}^{d_x}$, $\lambda > 0$.

Let $\boldsymbol{\theta} = \{\theta_w^{\mathsf{s}}, \theta_{y0}^{\mathsf{s}}, \theta_{y1}^{\mathsf{s}}, \theta_x^{\mathsf{s}}\}_{\mathsf{s}\in\mathcal{S}}$. Let $\mathcal{V}_w, \mathcal{V}_{y0}, \mathcal{V}_{y1}, \mathcal{V}_x$ be $1/(2\sqrt{m})$-packing of the unit $\|\cdot\|_2$-balls with cardinality at least $(2\sqrt{m})^{2B}, (2\sqrt{m})^{2B}, (2\sqrt{m})^{2B}, (2\sqrt{m})^{2Bd_x}$, respectively. Let $\mathcal{V}^{\mathsf{s}} = \delta(\mathcal{V}_w \times \mathcal{V}_{y0} \times \mathcal{V}_{y1} \times \mathcal{V}_x)$ and $\mathcal{V} = \mathcal{V}^{\mathsf{s}_1} \times \mathcal{V}^{\mathsf{s}_2} \times ... \times \mathcal{V}^{\mathsf{s}_m}$. We see that

$$|\mathcal{V}| \geq (2\sqrt{m})^{2mB(d_x+3)}.$$

In the following, we derive the minimax bound:

*Proof.* We have that

$$\|\boldsymbol{\theta}_1 - \boldsymbol{\theta}_2\|_2 = \sqrt{\sum_{\mathsf{s}\in\mathcal{S}}\sum_{c\in\mathcal{A}} \|(\theta_c^{\mathsf{s}})_1 - (\theta_c^{\mathsf{s}})_2\|_2^2} \geq \sqrt{\sum_{\mathsf{s}\in\mathcal{S}} 4\left(\frac{\delta}{2\sqrt{m}}\right)^2} = \delta.$$

The marginal distribution

$$p_{\boldsymbol{\theta}}(w,y,\boldsymbol{x}) = \int p_{\boldsymbol{\theta}}(w,y,\boldsymbol{x},\boldsymbol{z})d\boldsymbol{z} = \int p_{\boldsymbol{\theta}}(y|w,\boldsymbol{z})p_{\boldsymbol{\theta}}(w|\boldsymbol{z})p_{\boldsymbol{\theta}}(\boldsymbol{x}|\boldsymbol{z})p(\boldsymbol{z})d\boldsymbol{z}.$$

Moreover, we have that

$$D_{\mathrm{KL}}(p_{\boldsymbol{\theta}_1}^n \,\|\, p_{\boldsymbol{\theta}_2}^n) = \sum_{\mathsf{s}\in\mathcal{S}} D_{\mathrm{KL}}(p_{\boldsymbol{\theta}_1}^{n_\mathsf{s}} \,\|\, p_{\boldsymbol{\theta}_2}^{n_\mathsf{s}}).$$

We divide the proof into three parts **(I)**, **(II)**, and **(III)**:

**(I) The upper bound of $D_{\mathbf{KL}}(p_{\theta_1}^{n_\mathsf{s}} \,\|\, p_{\theta_2}^{n_\mathsf{s}})$**

Since the data is independent, we have that

$$D_{\mathrm{KL}}(p_{\boldsymbol{\theta}_1}^{n_\mathsf{s}} \,\|\, p_{\boldsymbol{\theta}_2}^{n_\mathsf{s}}) = n_\mathsf{s} D_{\mathrm{KL}}(p_{\boldsymbol{\theta}_1}^1 \,\|\, p_{\boldsymbol{\theta}_2}^1)$$

$$\leq n_\mathsf{s} \int D_{\mathrm{KL}}\Big( p_{\boldsymbol{\theta}_1}(y|w,\boldsymbol{z})p_{\boldsymbol{\theta}_1}(w|\boldsymbol{z})p_{\boldsymbol{\theta}_1}(\boldsymbol{x}|\boldsymbol{z})\Big\|p_{\boldsymbol{\theta}_2}(y|w,\boldsymbol{z}')p_{\boldsymbol{\theta}_2}(w|\boldsymbol{z}')p_{\boldsymbol{\theta}_2}(\boldsymbol{x}|\boldsymbol{z}')\Big) p(\boldsymbol{z})p(\boldsymbol{z}')d\boldsymbol{z}d\boldsymbol{z}'$$

$$= n_\mathsf{s} \int \Big[ p_{\boldsymbol{\theta}_1}(w=0|\boldsymbol{z})D_{\mathrm{KL}}\big[p_{\boldsymbol{\theta}_1}(y|w=0,\boldsymbol{z})\big\|p_{\boldsymbol{\theta}_2}(y|w=0,\boldsymbol{z}')\big]$$

$$+ p_{\boldsymbol{\theta}_1}(w=1|\boldsymbol{z})D_{\mathrm{KL}}\big[p_{\boldsymbol{\theta}_1}(y|w=1,\boldsymbol{z})\big\|p_{\boldsymbol{\theta}_2}(y|w=1,\boldsymbol{z}')\big]$$

$$+ D_{\mathrm{KL}}\big[p_{\boldsymbol{\theta}_1}(w|\boldsymbol{z})\big\|p_{\boldsymbol{\theta}_2}(w|\boldsymbol{z}')\big] + D_{\mathrm{KL}}\big[p_{\boldsymbol{\theta}_1}(\boldsymbol{x}|\boldsymbol{z})\big\|p_{\boldsymbol{\theta}_2}(\boldsymbol{x}|\boldsymbol{z}')\big] \Big] p(\boldsymbol{z})p(\boldsymbol{z}')d\boldsymbol{z}d\boldsymbol{z}'.$$

In the following, we find the upper bound of each component.

$\diamond$ *Upper bound of the first and second component*

$$p_{\boldsymbol{\theta}_1}(w=0|\boldsymbol{z})D_{\mathrm{KL}}\big[p_{\boldsymbol{\theta}_1}(y|w=0,\boldsymbol{z})\big\|p_{\boldsymbol{\theta}_2}(y|w=0,\boldsymbol{z}')\big]$$

$$\leq \frac{1}{2\sigma_y^2}\Big( \big((\theta_{y0}^\mathsf{s})_1 + \sum_{\mathsf{v}\in\mathcal{S}_{\backslash\mathsf{s}}} \lambda^{\mathsf{s},\mathsf{v}}(\theta_{y0}^\mathsf{v})_1\big)^\top \phi(\boldsymbol{z}) - \big((\theta_{y0}^\mathsf{s})_2 + \sum_{\mathsf{v}\in\mathcal{S}_{\backslash\mathsf{s}}} \lambda^{\mathsf{s},\mathsf{v}}(\theta_{y0}^\mathsf{v})_2\big)^\top \phi(\boldsymbol{z}')\Big)^2$$

$$\leq \frac{8B^2\delta^2(1+\sum_{\mathsf{v}\in\mathcal{S}_{\backslash\mathsf{s}}} \lambda^{\mathsf{s},\mathsf{v}})^2}{\sigma_y^2}.$$

Similarly, we also have

$$p_{\boldsymbol{\theta}_1}(w=1|\boldsymbol{z})D_{\mathrm{KL}}\big[p_{\boldsymbol{\theta}_1}(y|w=1,\boldsymbol{z})\big\|p_{\boldsymbol{\theta}_2}(y|w=1,\boldsymbol{z}')\big] \leq \frac{8B^2\delta^2(1+\sum_{\mathsf{v}\in\mathcal{S}_{\backslash\mathsf{s}}} \lambda^{\mathsf{s},\mathsf{v}})^2}{\sigma_y^2}.$$

$\diamond$ *Upper bound of the third component*

$$D_{\mathrm{KL}}\big[p_{\boldsymbol{\theta}_1}(w|\boldsymbol{z})\big\|p_{\boldsymbol{\theta}_2}(w|\boldsymbol{z}')\big]$$

$$= \varphi\Big(\big((\theta_w^\mathsf{s})_1 + \sum_{\mathsf{v}\in\mathcal{S}_{\backslash\mathsf{s}}} \lambda^{\mathsf{s},\mathsf{v}}(\theta_w^\mathsf{v})_1\big)^\top \phi(\boldsymbol{z})\Big) \log \frac{\varphi\Big(\big((\theta_w^\mathsf{s})_1 + \sum_{\mathsf{v}\in\mathcal{S}_{\backslash\mathsf{s}}} \lambda^{\mathsf{s},\mathsf{v}}(\theta_w^\mathsf{v})_1\big)^\top \phi(\boldsymbol{z})\Big)}{\varphi\Big(\big((\theta_w^\mathsf{s})_2 + \sum_{\mathsf{v}\in\mathcal{S}_{\backslash\mathsf{s}}} \lambda^{\mathsf{s},\mathsf{v}}(\theta_w^\mathsf{v})_2\big)^\top \phi(\boldsymbol{z}')\Big)}$$

$$+ \varphi\Big(-\big((\theta_w^\mathsf{s})_1 + \sum_{\mathsf{v}\in\mathcal{S}_{\backslash\mathsf{s}}} \lambda^{\mathsf{s},\mathsf{v}}(\theta_w^\mathsf{v})_1\big)^\top \phi(\boldsymbol{z})\Big) \log \frac{\varphi\Big(-\big((\theta_w^\mathsf{s})_1 + \sum_{\mathsf{v}\in\mathcal{S}_{\backslash\mathsf{s}}} \lambda^{\mathsf{s},\mathsf{v}}(\theta_w^\mathsf{v})_1\big)^\top \phi(\boldsymbol{z})\Big)}{\varphi\Big(-\big((\theta_w^\mathsf{s})_2 + \sum_{\mathsf{v}\in\mathcal{S}_{\backslash\mathsf{s}}} \lambda^{\mathsf{s},\mathsf{v}}(\theta_w^\mathsf{v})_2\big)^\top \phi(\boldsymbol{z}')\Big)}.$$

For the first component,

$$\varphi\Big(\big((\theta_w^\mathsf{s})_1 + \sum_{\mathsf{v}\in\mathcal{S}_{\backslash\mathsf{s}}} \lambda^{\mathsf{s},\mathsf{v}}(\theta_w^\mathsf{v})_1\big)^\top \phi(\mathbf{z})\Big) \log \frac{\varphi\Big(\big((\theta_w^\mathsf{s})_1 + \sum_{\mathsf{v}\in\mathcal{S}_{\backslash\mathsf{s}}} \lambda^{\mathsf{s},\mathsf{v}}(\theta_w^\mathsf{v})_1\big)^\top \phi(\mathbf{z})\Big)}{\varphi\Big(\big((\theta_w^\mathsf{s})_2 + \sum_{\mathsf{v}\in\mathcal{S}_{\backslash\mathsf{s}}} \lambda^{\mathsf{s},\mathsf{v}}(\theta_w^\mathsf{v})_2\big)^\top \phi(\mathbf{z}')\Big)}$$

$$\leq \Big| \log\big(1 + e^{-\big((\theta_w^\mathsf{s})_2 + \sum_{\mathsf{v}\in\mathcal{S}_{\backslash\mathsf{s}}} \lambda^{\mathsf{s},\mathsf{v}}(\theta_w^\mathsf{v})_2\big)^\top \phi(\mathbf{z})}\big) - \log\big(1 + e^{-\big((\theta_w^\mathsf{s})_1 + \sum_{\mathsf{v}\in\mathcal{S}_{\backslash\mathsf{s}}} \lambda^{\mathsf{s},\mathsf{v}}(\theta_w^\mathsf{v})_1\big)^\top \phi(\mathbf{z}')}\big) \Big|$$

$$\leq \left\| (\theta_w^{\mathsf{s}})_1 + \sum_{\mathsf{v}\in\mathcal{S}_{\backslash\mathsf{s}}} \lambda^{\mathsf{s},\mathsf{v}}(\theta_w^{\mathsf{v}})_1 \right\|_2 \|\phi(\mathbf{z})\|_2 + \left\| (\theta_w^{\mathsf{s}})_2 + \sum_{\mathsf{v}\in\mathcal{S}_{\backslash\mathsf{s}}} \lambda^{\mathsf{s},\mathsf{v}}(\theta_w^{\mathsf{v}})_2 \right\|_2 \|\phi(\mathbf{z}')\|_2$$

$$\leq \Big(\delta + \sum_{\mathsf{v}\in\mathcal{S}_{\backslash\mathsf{s}}} \lambda^{\mathsf{s},\mathsf{v}}\delta\Big)\|\phi(\mathbf{z})\|_2 + \Big(\delta + \sum_{\mathsf{v}\in\mathcal{S}_{\backslash\mathsf{s}}} \lambda^{\mathsf{s},\mathsf{v}}\delta\Big)\|\phi(\mathbf{z}')\|_2$$

$$\leq 4B\delta\Big(1 + \sum_{\mathsf{v}\in\mathcal{S}_{\backslash\mathsf{s}}} \lambda^{\mathsf{s},\mathsf{v}}\Big).$$

Similarly, we also have

$$\varphi\Big( -\Big((\theta_w^{\mathsf{s}})_1 + \sum_{\mathsf{v}\in\mathcal{S}_{\backslash\mathsf{s}}} \lambda^{\mathsf{s},\mathsf{v}}(\theta_w^{\mathsf{v}})_1\Big)^\top \phi(\mathbf{z})\Big) \log \frac{\varphi\Big( -\big((\theta_w^{\mathsf{s}})_1 + \sum_{\mathsf{v}\in\mathcal{S}_{\backslash\mathsf{s}}} \lambda^{\mathsf{s},\mathsf{v}}(\theta_w^{\mathsf{v}})_1\big)^\top \phi(\mathbf{z})\Big)}{\varphi\Big( -\big((\theta_w^{\mathsf{s}})_2 + \sum_{\mathsf{v}\in\mathcal{S}_{\backslash\mathsf{s}}} \lambda^{\mathsf{s},\mathsf{v}}(\theta_w^{\mathsf{v}})_2\big)^\top \phi(\mathbf{z}')\Big)}$$

$$\leq 4B\delta\Big(1 + \sum_{\mathsf{v}\in\mathcal{S}_{\backslash\mathsf{s}}} \lambda^{\mathsf{s},\mathsf{v}}\Big).$$

Thus,

$$D_{\mathrm{KL}}\big[p_{\boldsymbol{\theta}_1}(w|\boldsymbol{z})\big\|p_{\boldsymbol{\theta}_2}(w|\boldsymbol{z}')\big] \leq 8B\delta\Big(1 + \sum_{\mathsf{v}\in\mathcal{S}_{\backslash\mathsf{s}}} \lambda^{\mathsf{s},\mathsf{v}}\Big).$$

◇ *Upper bound of the fourth component*

$$D_{\mathrm{KL}}\big[p_{\boldsymbol{\theta}_1}(\boldsymbol{x}|\boldsymbol{z})\big\|p_{\boldsymbol{\theta}_2}(\boldsymbol{x}|\boldsymbol{z}')\big]$$

$$= \frac{1}{2\sigma_x^2}\left\| \Big((\theta_x^{\mathsf{s}})_1 + \sum_{\mathsf{v}\in\mathcal{S}_{\backslash\mathsf{s}}} \lambda^{\mathsf{s},\mathsf{v}}(\theta_x^{\mathsf{v}})_1\Big)^\top \phi(\boldsymbol{z}) - \Big((\theta_x^{\mathsf{s}})_2 + \sum_{\mathsf{v}\in\mathcal{S}_{\backslash\mathsf{s}}} \lambda^{\mathsf{s},\mathsf{v}}(\theta_x^{\mathsf{v}})_2\Big)^\top \phi(\boldsymbol{z}') \right\|_2^2$$

$$\leq \frac{1}{2\sigma_x^2}\left( \left\| \Big((\theta_x^{\mathsf{s}})_1 + \sum_{\mathsf{v}\in\mathcal{S}_{\backslash\mathsf{s}}} \lambda^{\mathsf{s},\mathsf{v}}(\theta_x^{\mathsf{v}})_1\Big)^\top \phi(\boldsymbol{z}) \right\|_2 + \left\| \Big((\theta_x^{\mathsf{s}})_2 + \sum_{\mathsf{v}\in\mathcal{S}_{\backslash\mathsf{s}}} \lambda^{\mathsf{s},\mathsf{v}}(\theta_x^{\mathsf{v}})_2\Big)^\top \phi(\boldsymbol{z}') \right\|_2 \right)^2$$

$$\leq \frac{8B^2\delta^2\Big(1 + \sum_{\mathsf{v}\in\mathcal{S}_{\backslash\mathsf{s}}} \lambda^{\mathsf{s},\mathsf{v}}\Big)^2}{\sigma_x^2}.$$

**(II) Combining the results**

From the above upper bound of each of the components, we obtain

$$D_{\mathrm{KL}}(p_{\boldsymbol{\theta}_1}^{n_{\mathsf{s}}} \,\|\, p_{\boldsymbol{\theta}_2}^{n_{\mathsf{s}}}) \leq n_{\mathsf{s}} \int \left[ \frac{16B^2\delta^2(1 + \sum_{\mathsf{v}\in\mathcal{S}_{\backslash\mathsf{s}}} \lambda^{\mathsf{s},\mathsf{v}})^2}{\sigma_y^2} + 8B\delta\Big(1 + \sum_{\mathsf{v}\in\mathcal{S}_{\backslash\mathsf{s}}} \lambda^{\mathsf{s},\mathsf{v}}\Big) \right.$$

$$\left. + \frac{8B^2\delta^2(1 + \sum_{\mathsf{v}\in\mathcal{S}_{\backslash\mathsf{s}}} \lambda^{\mathsf{s},\mathsf{v}})^2}{\sigma_x^2} \right] p(\boldsymbol{z})p(\boldsymbol{z}')d\boldsymbol{z}d\boldsymbol{z}'$$

$$= n_{\mathsf{s}}\left[ \Big(\frac{1}{\sigma_y^2} + \frac{1}{2\sigma_x^2}\Big)16B^2\delta^2\Big(1 + \sum_{\mathsf{v}\in\mathcal{S}_{\backslash\mathsf{s}}} \lambda^{\mathsf{s},\mathsf{v}}\Big)^2 + 8B\delta\Big(1 + \sum_{\mathsf{v}\in\mathcal{S}_{\backslash\mathsf{s}}} \lambda^{\mathsf{s},\mathsf{v}}\Big) \right].$$

**(III) The minimax lower bound**

We have that

$$D_{\mathrm{KL}}(p_{\boldsymbol{\theta}_1}^{n} \,\|\, p_{\boldsymbol{\theta}_2}^{n}) = \sum_{\mathsf{s}\in\mathcal{S}} D_{\mathrm{KL}}(p_{\boldsymbol{\theta}_1}^{n_{\mathsf{s}}} \,\|\, p_{\boldsymbol{\theta}_2}^{n_{\mathsf{s}}})$$

$$\leq \sum_{\mathsf{s}\in\mathcal{S}} n_{\mathsf{s}}\left[ \Big(\frac{1}{\sigma_y^2} + \frac{1}{2\sigma_x^2}\Big)16B^2\delta^2\Big(1 + \sum_{\mathsf{v}\in\mathcal{S}_{\backslash\mathsf{s}}} \lambda^{\mathsf{s},\mathsf{v}}\Big)^2 + 8B\delta\Big(1 + \sum_{\mathsf{v}\in\mathcal{S}_{\backslash\mathsf{s}}} \lambda^{\mathsf{s},\mathsf{v}}\Big) \right].$$

Consequently,

$$\inf_{\hat{\boldsymbol{\theta}}_n} \sup_{P\in\mathcal{P}} \mathbb{E}_P\left[ \|\hat{\boldsymbol{\theta}}_n - \boldsymbol{\theta}(P)\|_2 \right]$$

$$\geq \frac{\delta}{2}\left(1 - \frac{\sum_{\mathsf{s}\in\mathcal{S}} n_{\mathsf{s}}\left[\left(\frac{1}{\sigma_y^2} + \frac{1}{2\sigma_x^2}\right)16B^2\delta^2\left(1 + \sum_{\mathsf{v}\in\mathcal{S}_{\backslash\mathsf{s}}}\lambda^{\mathsf{s},\mathsf{v}}\right)^2 + 8B\delta\left(1 + \sum_{\mathsf{v}\in\mathcal{S}_{\backslash\mathsf{s}}}\lambda^{\mathsf{s},\mathsf{v}}\right)\right] + \log 2}{\log|\mathcal{V}|}\right)$$

$$\geq \frac{\delta}{2}\left(1 - \frac{\sum_{\mathsf{s}\in\mathcal{S}} n_{\mathsf{s}}\left[\left(\frac{1}{\sigma_y^2} + \frac{1}{2\sigma_x^2}\right)16B^2\delta^2\left(1 + \sum_{\mathsf{v}\in\mathcal{S}_{\backslash\mathsf{s}}}\lambda^{\mathsf{s},\mathsf{v}}\right)^2 + 8B\delta\left(1 + \sum_{\mathsf{v}\in\mathcal{S}_{\backslash\mathsf{s}}}\lambda^{\mathsf{s},\mathsf{v}}\right)\right] + \log 2}{2mB(d_x + 3)\log(2\sqrt{m})}\right).$$

We choose $\delta = \frac{\sqrt{mB(d_x+3)}\log(2\sqrt{m})}{4B\sum_{\mathsf{s}\in\mathcal{S}} n_{\mathsf{s}}\left(1 + \sum_{\mathsf{v}\in\mathcal{S}_{\backslash\mathsf{s}}}\lambda^{\mathsf{s},\mathsf{v}}\right)^2}$, then

$$1 - \frac{\sum_{\mathsf{s}\in\mathcal{S}} n_{\mathsf{s}}\left[\left(\frac{1}{\sigma_y^2} + \frac{1}{2\sigma_x^2}\right)16B^2\delta^2\left(1 + \sum_{\mathsf{v}\in\mathcal{S}_{\backslash\mathsf{s}}}\lambda^{\mathsf{s},\mathsf{v}}\right)^2 + 8B\delta\left(1 + \sum_{\mathsf{v}\in\mathcal{S}_{\backslash\mathsf{s}}}\lambda^{\mathsf{s},\mathsf{v}}\right)\right] + \log 2}{2mB(d_x + 3)\log(2\sqrt{m})}$$

$$\geq 1 - \left(\frac{1}{\sigma_y^2} + \frac{1}{2\sigma_x^2}\right)\frac{\log(2\sqrt{m})}{2\sum_{\mathsf{s}\in\mathcal{S}} n_{\mathsf{s}}\left(1 + \sum_{\mathsf{v}\in\mathcal{S}_{\backslash\mathsf{s}}}\lambda^{\mathsf{s},\mathsf{v}}\right)^2} - \frac{1}{\sqrt{mB(d_x+3)}} - \frac{1}{2mB(d_x+3)}$$

$$\geq 1 - \left(\frac{1}{\sigma_y^2} + \frac{1}{2\sigma_x^2}\right)\frac{\log(2\sqrt{m})}{2\sum_{\mathsf{s}\in\mathcal{S}} n_{\mathsf{s}}\left(1 + \sum_{\mathsf{v}\in\mathcal{S}_{\backslash\mathsf{s}}}\lambda^{\mathsf{s},\mathsf{v}}\right)^2} - \frac{1}{2} - \frac{1}{8}.$$

If $\sum_{\mathsf{s}\in\mathcal{S}} n_{\mathsf{s}}\left(1 + \sum_{\mathsf{v}\in\mathcal{S}_{\backslash\mathsf{s}}}\lambda^{\mathsf{s},\mathsf{v}}\right)^2 \geq 2\left(\frac{1}{\sigma_y^2} + \frac{1}{2\sigma_x^2}\right)\log(2\sqrt{m})$, then

$$\inf_{\hat{\theta}_n}\sup_{P\in\mathcal{P}}\mathbb{E}_P\left[\|\hat{\theta}_n - \theta(P)\|_2\right] \geq \frac{1}{2}\times\frac{\sqrt{mB(d_x+3)}\log(2\sqrt{m})}{4B\sum_{\mathsf{s}\in\mathcal{S}} n_{\mathsf{s}}\left(1 + \sum_{\mathsf{v}\in\mathcal{S}_{\backslash\mathsf{s}}}\lambda^{\mathsf{s},\mathsf{v}}\right)^2}\times\left(1 - \frac{1}{4} - \frac{1}{2} - \frac{1}{8}\right)$$

$$= \frac{\sqrt{m(d_x+3)}\log(2\sqrt{m})}{64\sqrt{B}\sum_{\mathsf{s}\in\mathcal{S}} n_{\mathsf{s}}\left(1 + \sum_{\mathsf{v}\in\mathcal{S}_{\backslash\mathsf{s}}}\lambda^{\mathsf{s},\mathsf{v}}\right)^2}.$$

This completes the proof. $\square$

## H  Proof of Lemma 2

We repeat Lemma 2 here for convenience:

**Lemma 2** (Without the presence of latent variables). *Let $\psi = \{\psi^{\mathsf{s}}\}_{\mathsf{s}=1}^m$, $\beta = \{\beta^{\mathsf{s}}\}_{\mathsf{s}=1}^m$ and $\hat{\psi}$, $\hat{\beta}$ be their estimates, respectively. Let $y_i^{\mathsf{s}}\in\mathbb{R}$. Then,*

**(i)** $\displaystyle\inf_{\hat{\psi}}\sup_{P\in\mathcal{P}}\mathbb{E}_P\left[\|\hat{\psi} - \psi(P)\|_2\right] \geq \frac{m\log(2\sqrt{m})}{256\sum_{\mathsf{s}\in\mathcal{S}} n_{\mathsf{s}}\left(1 + \sum_{\mathsf{v}\in\mathcal{S}, \mathsf{v}\neq\mathsf{s}}\gamma^{\mathsf{s},\mathsf{v}}\right)}$,

**(ii)** $\displaystyle\inf_{\hat{\beta}}\sup_{P\in\mathcal{P}}\mathbb{E}_P\left[\|\hat{\beta} - \beta(P)\|_2\right] \geq \frac{\sigma}{16\sqrt{2}}\sqrt{\frac{m\log(2\sqrt{m})}{B\sum_{\mathsf{s}\in\mathcal{S}} n_{\mathsf{s}}\left(1 + \sum_{\mathsf{v}\in\mathcal{S}, \mathsf{v}\neq\mathsf{s}}\eta^{\mathsf{s},\mathsf{v}}\right)^2}}.$

The proof of Lemma 2 is divided into two parts **(i)** and **(ii)**. We compute them separately:

### H.1  Proof of Part (i)

We summarize the model as follows

$$w^{\mathsf{s}} \sim \mathsf{Bern}\left(\varphi\left(\left(\psi^{\mathsf{s}} + \sum_{\mathsf{v}\in\mathcal{S}_{\backslash\mathsf{s}}}\gamma^{\mathsf{s},\mathsf{v}}\psi^{\mathsf{v}}\right)^{\top}\phi(\boldsymbol{x}^{\mathsf{s}})\right)\right).$$

Let $\boldsymbol{\psi} = \{\psi^{\mathsf{s}}\}_{\mathsf{s}\in\mathcal{S}}$. Let $\mathcal{V}_{\mathsf{s}}$ be $1/(2\sqrt{m})$-packing of the unit $\|\cdot\|_2$-balls with cardinality at least $(2\sqrt{m})^{2B}$. We now choose a set $\mathcal{V} = \delta(\mathcal{V}_{\mathsf{s}_1} \times \mathcal{V}_{\mathsf{s}_2} \times ... \times \mathcal{V}_{\mathsf{s}_m})$. We see that

$$|\mathcal{V}| \geq (2\sqrt{m})^{2mB}.$$

*Proof.* We have that

$$\|\boldsymbol{\psi}_1 - \boldsymbol{\psi}_2\|_2 = \sqrt{\sum_{\mathsf{s}\in\mathcal{S}} \|\psi_1^{\mathsf{s}} - \psi_2^{\mathsf{s}}\|_2^2} \geq \delta/2.$$

Moreover,

$$D_{\mathrm{KL}}(p_{\boldsymbol{\psi}_1}^n \,\|\, p_{\boldsymbol{\psi}_2}^n) = \sum_{\mathsf{s}\in\mathcal{S}} D_{\mathrm{KL}}(p_{\boldsymbol{\psi}_1}^{n_{\mathsf{s}}} \,\|\, p_{\boldsymbol{\psi}_2}^{n_{\mathsf{s}}}).$$

We first find upper bound of $D_{\mathrm{KL}}(p_{\boldsymbol{\psi}_1}^{n_{\mathsf{s}}} \,\|\, p_{\boldsymbol{\psi}_2}^{n_{\mathsf{s}}})$. Since the data is independent, we have that

$$D_{\mathrm{KL}}(p_{\boldsymbol{\psi}_1}^{n_{\mathsf{s}}} \,\|\, p_{\boldsymbol{\psi}_2}^{n_{\mathsf{s}}}) = n_{\mathsf{s}} D_{\mathrm{KL}}(p_{\boldsymbol{\psi}_1}^1 \,\|\, p_{\boldsymbol{\psi}_2}^1)$$

$$= n_{\mathsf{s}}\left[ \varphi\Big(\big(\psi_1^{\mathsf{s}} + \sum_{\mathsf{v}\in\mathcal{S}_{\backslash\mathsf{s}}} \gamma^{\mathsf{s},\mathsf{v}}\psi_1^{\mathsf{v}}\big)^{\top}\phi(\boldsymbol{x}^{\mathsf{s}})\Big) \log \frac{\varphi\Big(\big(\psi_1^{\mathsf{s}} + \sum_{\mathsf{v}\in\mathcal{S}_{\backslash\mathsf{s}}} \gamma^{\mathsf{s},\mathsf{v}}\psi_1^{\mathsf{v}}\big)^{\top}\phi(\boldsymbol{x}^{\mathsf{s}})\Big)}{\varphi\Big(\big(\psi_2^{\mathsf{s}} + \sum_{\mathsf{v}\in\mathcal{S}_{\backslash\mathsf{s}}} \gamma^{\mathsf{s},\mathsf{v}}\psi_2^{\mathsf{v}}\big)^{\top}\phi(\boldsymbol{x}^{\mathsf{s}})\Big)}$$

$$+ \varphi\Big(-\big(\psi_1^{\mathsf{s}} + \sum_{\mathsf{v}\in\mathcal{S}_{\backslash\mathsf{s}}} \gamma^{\mathsf{s},\mathsf{v}}\psi_1^{\mathsf{v}}\big)^{\top}\phi(\boldsymbol{x}^{\mathsf{s}})\Big) \log \frac{\varphi\Big(-\big(\psi_1^{\mathsf{s}} + \sum_{\mathsf{v}\in\mathcal{S}_{\backslash\mathsf{s}}} \gamma^{\mathsf{s},\mathsf{v}}\psi_1^{\mathsf{v}}\big)^{\top}\phi(\boldsymbol{x}^{\mathsf{s}})\Big)}{\varphi\Big(-\big(\psi_2^{\mathsf{s}} + \sum_{\mathsf{v}\in\mathcal{S}_{\backslash\mathsf{s}}} \gamma^{\mathsf{s},\mathsf{v}}\psi_2^{\mathsf{v}}\big)^{\top}\phi(\boldsymbol{x}^{\mathsf{s}})\Big)} \right].$$

The first component:

$$\varphi\Big(\big(\psi_1^{\mathsf{s}} + \sum_{\mathsf{v}\in\mathcal{S}_{\backslash\mathsf{s}}} \gamma^{\mathsf{s},\mathsf{v}}\psi_1^{\mathsf{v}}\big)^{\top}\phi(\mathbf{x}^{\mathsf{s}})\Big) \log \frac{\varphi\Big(\big(\psi_1^{\mathsf{s}} + \sum_{\mathsf{v}\in\mathcal{S}_{\backslash\mathsf{s}}} \gamma^{\mathsf{s},\mathsf{v}}\psi_1^{\mathsf{v}}\big)^{\top}\phi(\mathbf{x}^{\mathsf{s}})\Big)}{\varphi\Big(\big(\psi_2^{\mathsf{s}} + \sum_{\mathsf{v}\in\mathcal{S}_{\backslash\mathsf{s}}} \gamma^{\mathsf{s},\mathsf{v}}\psi_2^{\mathsf{v}}\big)^{\top}\phi(\mathbf{x}^{\mathsf{s}})\Big)}$$

$$\leq \left| \log\Big(1 + e^{-\big(\psi_2^{\mathsf{s}} + \sum_{\mathsf{v}\in\mathcal{S}_{\backslash\mathsf{s}}} \gamma^{\mathsf{s},\mathsf{v}}\psi_2^{\mathsf{v}}\big)^{\top}\phi(\mathbf{x}^{\mathsf{s}})}\Big) - \log\Big(1 + e^{-\big(\psi_1^{\mathsf{s}} + \sum_{\mathsf{v}\in\mathcal{S}_{\backslash\mathsf{s}}} \gamma^{\mathsf{s},\mathsf{v}}\psi_1^{\mathsf{v}}\big)^{\top}\phi(\mathbf{x}^{\mathsf{s}})}\Big) \right|$$

$$\overset{(\star)}{\leq} \left| \big(\psi_2^{\mathsf{s}} + \sum_{\mathsf{v}\in\mathcal{S}_{\backslash\mathsf{s}}} \gamma^{\mathsf{s},\mathsf{v}}\psi_2^{\mathsf{v}}\big)^{\top}\phi(\mathbf{x}^{\mathsf{s}}) - \big(\psi_1^{\mathsf{s}} + \sum_{\mathsf{v}\in\mathcal{S}_{\backslash\mathsf{s}}} \gamma^{\mathsf{s},\mathsf{v}}\psi_1^{\mathsf{v}}\big)^{\top}\phi(\mathbf{x}^{\mathsf{s}}) \right|$$

$$\leq 4B\delta\Big(1 + \sum_{\mathsf{v}\in\mathcal{S}_{\backslash\mathsf{s}}} \gamma^{\mathsf{s},\mathsf{v}}\Big),$$

where $(\star)$ follows from the fact that the SoftPlus function $\log(1 + e^x)$ is 1-Lipschitz. In particular,

$$\big| \log(1 + e^{x_1}) - \log(1 + e^{x_2}) \big| = \left| \int_{x_1}^{x_2} \frac{e^x}{1 + e^x} dx \right| \leq \left| \int_{x_1}^{x_2} 1 dx \right| = |x_1 - x_2|.$$

Similarly, for the second component, we also have

$$\varphi\Big(-\big(\psi_1^{\mathsf{s}} + \sum_{\mathsf{v}\in\mathcal{S}_{\backslash\mathsf{s}}} \gamma^{\mathsf{s},\mathsf{v}}\psi_1^{\mathsf{v}}\big)^{\top}\phi(\boldsymbol{x}^{\mathsf{s}})\Big) \log \frac{\varphi\Big(-\big(\psi_1^{\mathsf{s}} + \sum_{\mathsf{v}\in\mathcal{S}_{\backslash\mathsf{s}}} \gamma^{\mathsf{s},\mathsf{v}}\psi_1^{\mathsf{v}}\big)^{\top}\phi(\boldsymbol{x}^{\mathsf{s}})\Big)}{\varphi\Big(-\big(\psi_2^{\mathsf{s}} + \sum_{\mathsf{v}\in\mathcal{S}_{\backslash\mathsf{s}}} \gamma^{\mathsf{s},\mathsf{v}}\psi_2^{\mathsf{v}}\big)^{\top}\phi(\boldsymbol{x}^{\mathsf{s}})\Big)}$$

$$\leq 4B\delta\Big(1 + \sum_{\mathsf{v}\in\mathcal{S}_{\backslash\mathsf{s}}} \gamma^{\mathsf{s},\mathsf{v}}\Big).$$

Thus,

$$D_{\mathrm{KL}}(p_{\boldsymbol{\psi}_1}^{n_{\mathsf{s}}} \,\|\, p_{\boldsymbol{\psi}_2}^{n_{\mathsf{s}}}) \leq 8B\delta\Big(1 + \sum_{\mathsf{v}\in\mathcal{S}_{\backslash\mathsf{s}}} \gamma^{\mathsf{s},\mathsf{v}}\Big) n_{\mathsf{s}}.$$

Consequently,

$$D_{\mathrm{KL}}(p^n_{\boldsymbol{\psi}_1} \,\|\, p^n_{\boldsymbol{\psi}_2}) \le 8B\delta \sum_{\mathsf{s}\in\mathcal{S}} n_{\mathsf{s}}\Big(1 + \sum_{\mathsf{v}\in\mathcal{S}_{\setminus\mathsf{s}}} \gamma^{\mathsf{s},\mathsf{v}}\Big).$$

So, we have that

$$\inf_{\hat{\boldsymbol{\psi}}_n} \sup_{P\in\mathcal{P}} \mathbb{E}_P\Big[\|\hat{\boldsymbol{\psi}}_n - \boldsymbol{\psi}(P)\|_2\Big] \ge \frac{\delta}{4}\left(1 - \frac{8B\delta \sum_{\mathsf{s}\in\mathcal{S}} n_{\mathsf{s}}\big(1 + \sum_{\mathsf{v}\in\mathcal{S}_{\setminus\mathsf{s}}} \gamma^{\mathsf{s},\mathsf{v}}\big) + \log 2}{\log|\mathcal{V}|}\right)$$

$$\ge \frac{\delta}{4}\left(1 - \frac{8B\delta \sum_{\mathsf{s}\in\mathcal{S}} n_{\mathsf{s}}\big(1 + \sum_{\mathsf{v}\in\mathcal{S}_{\setminus\mathsf{s}}} \gamma^{\mathsf{s},\mathsf{v}}\big) + \log 2}{2mB\log(2\sqrt{m})}\right).$$

We choose $\delta = \frac{m\log(2\sqrt{m})}{16\sum_{\mathsf{s}\in\mathcal{S}} n_{\mathsf{s}}\big(1+\sum_{\mathsf{v}\in\mathcal{S}_{\setminus\mathsf{s}}} \gamma^{\mathsf{s},\mathsf{v}}\big)}$, then

$$1 - \frac{8B\delta \sum_{\mathsf{s}\in\mathcal{S}} n_{\mathsf{s}}\big(1 + \sum_{\mathsf{v}\in\mathcal{S}_{\setminus\mathsf{s}}} \gamma^{\mathsf{s},\mathsf{v}}\big) + \log 2}{2mB\log(2\sqrt{m})} \ge \frac{1}{4}.$$

Thus,

$$\inf_{\hat{\boldsymbol{\psi}}_n} \sup_{P\in\mathcal{P}} \mathbb{E}_P\Big[\|\hat{\boldsymbol{\psi}}_n - \boldsymbol{\psi}(P)\|_2\Big] \ge \frac{1}{4} \times \frac{mB\log(2\sqrt{m})}{16B\sum_{\mathsf{s}\in\mathcal{S}} n_{\mathsf{s}}\big(1 + \sum_{\mathsf{v}\in\mathcal{S}_{\setminus\mathsf{s}}} \gamma^{\mathsf{s},\mathsf{v}}\big)} \times \frac{1}{4}$$

$$= \frac{m\log(2\sqrt{m})}{256\sum_{\mathsf{s}\in\mathcal{S}} n_{\mathsf{s}}\big(1 + \sum_{\mathsf{v}\in\mathcal{S}_{\setminus\mathsf{s}}} \gamma^{\mathsf{s},\mathsf{v}}\big)}.$$

This completes the proof of part (i). $\qquad\square$

### H.2   Proof of Part (ii)

*Proof.* We summarize the model as follows

$$y^{\mathsf{s}} = \left((1 - w^{\mathsf{s}})\big(\beta_0^{\mathsf{s}} + \sum_{\mathsf{v}\in\mathcal{S}_{\setminus\mathsf{s}}} \eta^{\mathsf{s},\mathsf{v}}\beta_0^{\mathsf{v}}\big) + w^{\mathsf{s}}\big(\beta_1^{\mathsf{s}} + \sum_{\mathsf{v}\in\mathcal{S}_{\setminus\mathsf{s}}} \eta^{\mathsf{s},\mathsf{v}}\beta_1^{\mathsf{v}}\big)\right)^{\top}\phi(\mathbf{x}^{\mathsf{s}}) + \epsilon_{\mathsf{s}}, \qquad \epsilon_{\mathsf{s}} \sim \mathsf{N}(0, \sigma^2).$$

Let $\boldsymbol{\beta} = \{\beta_0^{\mathsf{s}}, \beta_1^{\mathsf{s}}\}_{\mathsf{s}\in\mathcal{S}}$. Let $\mathcal{V}_{0\mathsf{s}}$ and $\mathcal{V}_{1\mathsf{s}}$ be $1/(2\sqrt{m})$-packing of the unit $\|\cdot\|_2$-balls with cardinality at least $(2\sqrt{m})^{2B}$. Let $\mathcal{V}_{\mathsf{s}} = \mathcal{V}_{0\mathsf{s}} \times \mathcal{V}_{1\mathsf{s}}$. We now choose a set $\mathcal{V} = \delta(\mathcal{V}_{\mathsf{s}_1} \times \mathcal{V}_{\mathsf{s}_2} \times ... \times \mathcal{V}_{\mathsf{s}_m})$. We see that

$$|\mathcal{V}| \ge (2\sqrt{m})^{4mB}.$$

We have that

$$\|\boldsymbol{\beta}_1 - \boldsymbol{\beta}_2\|_2 = \sqrt{\sum_{\mathsf{s}\in\mathcal{S}}\Big(\|(\beta_0^{\mathsf{s}})_1 - (\beta_0^{\mathsf{s}})_2\|_2^2 + \|(\beta_1^{\mathsf{s}})_1 - (\beta_1^{\mathsf{s}})_2\|_2^2\Big)} \ge \delta/\sqrt{2}.$$

Moreover,

$$D_{\mathrm{KL}}(p^n_{\boldsymbol{\beta}_1} \,\|\, p^n_{\boldsymbol{\beta}_2}) = \sum_{\mathsf{s}\in\mathcal{S}} D_{\mathrm{KL}}(p^{n_{\mathsf{s}}}_{\boldsymbol{\beta}_1} \,\|\, p^{n_{\mathsf{s}}}_{\boldsymbol{\beta}_2}) = \sum_{\mathsf{s}\in\mathcal{S}} n_{\mathsf{s}} D_{\mathrm{KL}}(p^1_{\boldsymbol{\beta}_1} \,\|\, p^1_{\boldsymbol{\beta}_2}).$$

In addition,

$$D_{\mathrm{KL}}(p^1_{\boldsymbol{\beta}_1} \,\|\, p^1_{\boldsymbol{\beta}_2})$$
$$= \frac{1}{2\sigma^2}\Bigg(\Big((1 - w^{\mathsf{s}})\big((\beta_0^{\mathsf{s}})_1 + \sum_{\mathsf{v}\in\mathcal{S}_{\setminus\mathsf{s}}} \eta^{\mathsf{s},\mathsf{v}}(\beta_0^{\mathsf{v}})_1\big) + w^{\mathsf{s}}\big((\beta_1^{\mathsf{s}})_1 + \sum_{\mathsf{v}\in\mathcal{S}_{\setminus\mathsf{s}}} \eta^{\mathsf{s},\mathsf{v}}(\beta_1^{\mathsf{v}})_1\big)\Big)^{\top}\phi(\mathbf{x}^{\mathsf{s}})$$
$$- \Big((1 - w^{\mathsf{s}})\big((\beta_0^{\mathsf{s}})_2 + \sum_{\mathsf{v}\in\mathcal{S}_{\setminus\mathsf{s}}} \eta^{\mathsf{s},\mathsf{v}}(\beta_0^{\mathsf{v}})_2\big) + w^{\mathsf{s}}\big((\beta_1^{\mathsf{s}})_2 + \sum_{\mathsf{v}\in\mathcal{S}_{\setminus\mathsf{s}}} \eta^{\mathsf{s},\mathsf{v}}(\beta_1^{\mathsf{v}})_2\big)\Big)^{\top}\phi(\mathbf{x}^{\mathsf{s}})\Bigg)^2$$

$$\leq \frac{1}{2\sigma^2}\left(\left((1-w^{\mathsf{s}})(2\delta + \sum_{\mathsf{v}\in\mathcal{S}_{\backslash\mathsf{s}}}\eta^{\mathsf{s},\mathsf{v}}2\delta) + w^{\mathsf{s}}(2\delta + \sum_{\mathsf{v}\in\mathcal{S}_{\backslash\mathsf{s}}}\eta^{\mathsf{s},\mathsf{v}}2\delta)\right)\|\phi(\mathbf{x}^{\mathsf{s}})\|_2\right)^2$$

$$\leq \frac{8B^2\delta^2}{\sigma^2}\left(1 + \sum_{\mathsf{v}\in\mathcal{S}_{\backslash\mathsf{s}}}\eta^{\mathsf{s},\mathsf{v}}\right)^2,$$

Thus,

$$D_{\mathrm{KL}}(p^n_{\boldsymbol{\beta}_1}\,\|\,p^n_{\boldsymbol{\beta}_2}) \leq \frac{8B^2\delta^2}{\sigma^2}\sum_{\mathsf{s}\in\mathcal{S}}n_{\mathsf{s}}\left(1 + \sum_{\mathsf{v}\in\mathcal{S}_{\backslash\mathsf{s}}}\eta^{\mathsf{s},\mathsf{v}}\right)^2.$$

Consequently,

$$\inf_{\hat{\boldsymbol{\beta}}_n}\sup_{P\in\mathcal{P}}\mathbb{E}_P\left[\|\hat{\boldsymbol{\beta}}_n - \boldsymbol{\beta}(P)\|_2\right] \geq \frac{\delta}{2\sqrt{2}}\left(1 - \frac{\frac{8B^2\delta^2}{\sigma^2}\sum_{\mathsf{s}\in\mathcal{S}}n_{\mathsf{s}}\left(1 + \sum_{\mathsf{v}\in\mathcal{S}_{\backslash\mathsf{s}}}\eta^{\mathsf{s},\mathsf{v}}\right)^2 + \log 2}{\log|\mathcal{V}|}\right)$$

$$\geq \frac{\delta}{2\sqrt{2}}\left(1 - \frac{\frac{8B^2\delta^2}{\sigma^2}\sum_{\mathsf{s}\in\mathcal{S}}n_{\mathsf{s}}\left(1 + \sum_{\mathsf{v}\in\mathcal{S}_{\backslash\mathsf{s}}}\eta^{\mathsf{s},\mathsf{v}}\right)^2 + \log 2}{4mB\log(2\sqrt{m})}\right).$$

We choose $\delta^2 = \dfrac{mB\log(2\sqrt{m})}{4\frac{B^2}{\sigma^2}\sum_{\mathsf{s}\in\mathcal{S}}n_{\mathsf{s}}\left(1 + \sum_{\mathsf{v}\in\mathcal{S}_{\backslash\mathsf{s}}}\eta^{\mathsf{s},\mathsf{v}}\right)^2}$, then

$$1 - \frac{\frac{8B^2\delta^2}{\sigma^2}\sum_{\mathsf{s}\in\mathcal{S}}n_{\mathsf{s}}\left(1 + \sum_{\mathsf{v}\in\mathcal{S}_{\backslash\mathsf{s}}}\eta^{\mathsf{s},\mathsf{v}}\right)^2 + \log 2}{4mB\log(2\sqrt{m})} = 1 - \frac{2mB\log(2\sqrt{m}) + \log 2}{4mB\log(2\sqrt{m})} \geq \frac{1}{4}.$$

Thus,

$$\inf_{\hat{\boldsymbol{\beta}}_n}\sup_{P\in\mathcal{P}}\mathbb{E}_P\left[\|\hat{\boldsymbol{\beta}}_n - \boldsymbol{\beta}(P)\|_2\right] \geq \frac{1}{2\sqrt{2}}\sqrt{\frac{4mB\log(2\sqrt{m})}{2\frac{8B^2}{\sigma^2}\sum_{\mathsf{s}\in\mathcal{S}}n_{\mathsf{s}}\left(1 + \sum_{\mathsf{v}\in\mathcal{S}_{\backslash\mathsf{s}}}\eta^{\mathsf{s},\mathsf{v}}\right)^2}} \times \frac{1}{4}$$

$$= \frac{\sigma}{16\sqrt{2}}\sqrt{\frac{m\log(2\sqrt{m})}{B\sum_{\mathsf{s}\in\mathcal{S}}n_{\mathsf{s}}\left(1 + \sum_{\mathsf{v}\in\mathcal{S}_{\backslash\mathsf{s}}}\eta^{\mathsf{s},\mathsf{v}}\right)^2}}.$$

This completes the proof of part (ii). $\qquad\square$

# I  Further cases of the minimax lower bounds

In Lemma 1 and 2, we have presented the minimax lower bounds when $y_i^{\mathsf{s}} \in \mathbb{R}$ and $\boldsymbol{x}_i^{\mathsf{s}} \in \mathbb{R}^{d_x}$. Here, we briefly describe the other cases.

## I.1  Further cases of Lemma 1

In this section, we further detail the lower bound for binary outcomes and binary proxy variables. In this case, we need to re-derive the upper bound of

$$p_{\boldsymbol{\theta}_1}(w=j|\boldsymbol{z})D_{\mathrm{KL}}\big[p_{\boldsymbol{\theta}_1}(y|w=j,\boldsymbol{z})\big\|p_{\boldsymbol{\theta}_2}(y|w=j,\boldsymbol{z}')\big] \quad \text{and} \quad D_{\mathrm{KL}}\big[p_{\boldsymbol{\theta}_1}(\boldsymbol{x}|\boldsymbol{z})\big\|p_{\boldsymbol{\theta}_2}(\boldsymbol{x}|\boldsymbol{z}')\big],$$

where $j = 1, 2$. Using similar derivations as before for the quantity $D_{\mathrm{KL}}\big[p_{\boldsymbol{\theta}_1}(w|\boldsymbol{z})\big\|p_{\boldsymbol{\theta}_2}(w|\boldsymbol{z}')\big]$, we have that

$$p_{\boldsymbol{\theta}_1}(w=j|\boldsymbol{z})D_{\mathrm{KL}}\big[p_{\boldsymbol{\theta}_1}(y|w=j,\boldsymbol{z})\big\|p_{\boldsymbol{\theta}_2}(y|w=j,\boldsymbol{z}')\big] \leq 8B\delta\left(1 + \sum_{\mathsf{v}\in\mathcal{S}_{\backslash\mathsf{s}}}\lambda^{\mathsf{s},\mathsf{v}}\right),$$

and

$$D_{\mathrm{KL}}\big[p_{\boldsymbol{\theta}_1}(\boldsymbol{x}|\boldsymbol{z})\big\|p_{\boldsymbol{\theta}_2}(\boldsymbol{x}|\boldsymbol{z}')\big] \leq d_x 8B\delta\left(1 + \sum_{\mathsf{v}\in\mathcal{S}_{\backslash\mathsf{s}}}\lambda^{\mathsf{s},\mathsf{v}}\right).$$

Combining the results, we have

$$D_{\mathrm{KL}}(p_{\boldsymbol{\theta}_1}^n \| p_{\boldsymbol{\theta}_2}^n) = \sum_{\mathsf{s} \in \mathcal{S}} D_{\mathrm{KL}}(p_{\boldsymbol{\theta}_1}^{n_\mathsf{s}} \| p_{\boldsymbol{\theta}_2}^{n_\mathsf{s}}) \leq \sum_{\mathsf{s} \in \mathcal{S}} n_\mathsf{s} 8(d_x + 3) B \delta \Big( 1 + \sum_{\mathsf{v} \in \mathcal{S}_{\backslash \mathsf{s}}} \lambda^{\mathsf{s},\mathsf{v}} \Big).$$

Consequently, we have that

$$\inf_{\hat{\boldsymbol{\theta}}_n} \sup_{P \in \mathcal{P}} \mathbb{E}_P \Big[ \| \hat{\boldsymbol{\theta}}_n - \boldsymbol{\theta}(P) \|_2 \Big] \geq \frac{\delta}{2} \left( 1 - \frac{\sum_{\mathsf{s} \in \mathcal{S}} n_\mathsf{s} 8(d_x + 3) B \delta \Big( 1 + \sum_{\mathsf{v} \in \mathcal{S}_{\backslash \mathsf{s}}} \lambda^{\mathsf{s},\mathsf{v}} \Big) + \log 2}{2mB(d_x + 3) \log(2\sqrt{m})} \right).$$

We choose $\delta = \dfrac{m \log(2\sqrt{m})}{8 \sum_{\mathsf{s} \in \mathcal{S}} n_\mathsf{s} \Big( 1 + \sum_{\mathsf{v} \in \mathcal{S}_{\backslash \mathsf{s}}} \lambda^{\mathsf{s},\mathsf{v}} \Big)}$, then

$$1 - \frac{\sum_{\mathsf{s} \in \mathcal{S}} n_\mathsf{s} 8(d_x + 3) B \delta \Big( 1 + \sum_{\mathsf{v} \in \mathcal{S}_{\backslash \mathsf{s}}} \lambda^{\mathsf{s},\mathsf{v}} \Big) + \log 2}{2mB(d_x + 3) \log(2\sqrt{m})} \geq \frac{3}{8}.$$

Thus,

$$\boxed{\inf_{\hat{\boldsymbol{\theta}}_n} \sup_{P \in \mathcal{P}} \mathbb{E}_P \Big[ \| \hat{\boldsymbol{\theta}}_n - \boldsymbol{\theta}(P) \|_2 \Big] \geq \frac{3mB \log(2\sqrt{m})}{128 \sum_{\mathsf{s} \in \mathcal{S}} n_\mathsf{s} B \Big( 1 + \sum_{\mathsf{v} \in \mathcal{S}_{\backslash \mathsf{s}}} \lambda^{\mathsf{s},\mathsf{v}} \Big)}.}$$

**Remark 1.** *Note that the derivation in this Section and in Section* H.1 *give us enough tools to compute the minimax lower bounds for any further case, i.e., any combination of the outcomes and proxy variables (binary or continuous). The key is to initially find the upper bound of* $D_{KL}(p_{\boldsymbol{\theta}_1}^n \| p_{\boldsymbol{\theta}_2}^n)$ *based on the constructed packing. Then, using Fano's method to obtain the minimax lower bounds.*

### I.2 Further cases of Lemma 2

Note that the lower bound of Lemma 2, part **(i)** has only one case since we only focus on binary treatment, and it is presented in the main text. For part **(ii)**, consider $y_i^\mathsf{s} \in \{0, 1\}$, then the model of the outcomes would follow a Bernoulli distribution. Reusing the scheme in Section H.2, we need to find the new upper bound of $D_{\mathrm{KL}}(p_{\boldsymbol{\beta}_1}^n \| p_{\boldsymbol{\beta}_2}^n)$. In particular,

$$D_{\mathrm{KL}}(p_{\boldsymbol{\beta}_1}^n \| p_{\boldsymbol{\beta}_2}^n) = \sum_{\mathsf{s} \in \mathcal{S}} n_\mathsf{s} \left[ \varphi(v_1) \log \frac{\varphi(v_1)}{\varphi(v_2)} + \varphi(-v_1) \log \frac{\varphi(-v_1)}{\varphi(-v_2)} \right],$$

where $v_j = \Big( (1 - w^\mathsf{s})\big( (\beta_0^\mathsf{s})_j + \sum_{\mathsf{v} \in \mathcal{S}_{\backslash \mathsf{s}}} \eta^{\mathsf{s},\mathsf{v}} (\beta_0^\mathsf{v})_j \big) + w^\mathsf{s} \big( (\beta_1^\mathsf{s})_j + \sum_{\mathsf{v} \in \mathcal{S}_{\backslash \mathsf{s}}} \eta^{\mathsf{s},\mathsf{v}} (\beta_1^\mathsf{v})_j \big) \Big)^\top \phi(\mathbf{x}^\mathsf{s})$. We have that

$$\varphi(v_1) \log \frac{\varphi(v_1)}{\varphi(v_2)} \leq \Big\| (1 - w^\mathsf{s})\big( (\beta_0^\mathsf{s})_1 - (\beta_0^\mathsf{s})_2 + \sum_{\mathsf{v} \in \mathcal{S}_{\backslash \mathsf{s}}} \eta^{\mathsf{s},\mathsf{v}}[(\beta_0^\mathsf{v})_1 - (\beta_0^\mathsf{v})_2] \big)$$

$$+ w^\mathsf{s}\big( (\beta_1^\mathsf{s})_1 - (\beta_1^\mathsf{s})_2 + \sum_{\mathsf{v} \in \mathcal{S}_{\backslash \mathsf{s}}} \eta^{\mathsf{s},\mathsf{v}}[(\beta_1^\mathsf{v})_1 - (\beta_1^\mathsf{v})_2] \big) \Big\|_2 \| \phi(\mathbf{x}^\mathsf{s}) \|_2$$

$$\leq 4B\delta \Big( 1 + \sum_{\mathsf{v} \in \mathcal{S}_{\backslash \mathsf{s}}} \gamma^{\mathsf{s},\mathsf{v}} \Big),$$

Similarly, $\varphi(-v_1) \log \frac{\varphi(-v_1)}{\varphi(-v_2)} \leq 4B\delta \Big( 1 + \sum_{\mathsf{v} \in \mathcal{S}_{\backslash \mathsf{s}}} \gamma^{\mathsf{s},\mathsf{v}} \Big)$. Hence,

$$D_{\mathrm{KL}}(p_{\boldsymbol{\beta}_1}^n \| p_{\boldsymbol{\beta}_2}^n) \leq 8B\delta \sum_{\mathsf{s} \in \mathcal{S}} n_\mathsf{s} \Big( 1 + \sum_{\mathsf{v} \in \mathcal{S}_{\backslash \mathsf{s}}} \eta^{\mathsf{s},\mathsf{v}} \Big).$$

Using similar technique in Section H.2, we obtain

$$\boxed{\inf_{\hat{\boldsymbol{\beta}}_n} \sup_{P \in \mathcal{P}} \mathbb{E}_P \Big[ \| \hat{\boldsymbol{\beta}}_n - \boldsymbol{\beta}(P) \|_2 \Big] \geq \frac{m \log(2\sqrt{m})}{32\sqrt{2} \sum_{\mathsf{s} \in \mathcal{S}} n_\mathsf{s} \Big( 1 + \sum_{\mathsf{v} \in \mathcal{S}_{\backslash \mathsf{s}}} \eta^{\mathsf{s},\mathsf{v}} \Big)}.}$$

We observe that the lower bound is similar to that of Lemma 2, part **(i)** since they are both lower bounds of a binary response variable. The constant in this bound is larger $(1/(32\sqrt{2}))$ than that of Lemma 2, part **(i)** $(1/256)$. This is expected since there are more parameters in this model, i.e., $\{\beta_0^\mathsf{s}, \beta_1^\mathsf{s}\}_{\mathsf{s} \in \mathcal{S}}$, as compared to the model in Lemma 2, part **(i)** $(\{\psi^\mathsf{s}\}_{\mathsf{s} \in \mathcal{S}})$.

## J    Description of IHDP data

This section describe details of the IHDP data, which was skipped in the main text due to limited space.

The Infant Health and Development Program (IHDP) is a randomized study on the impact of specialist visits (the treatment) on the cognitive development of children (the outcome). The dataset consists of 747 records with 25 covariates describing properties of the children and their mothers. The treatment group includes children who received specialist visits and control group includes children who did not receive. Further details are presented in Appendix. For each child, a treated and a control outcome are simulated using the numerical schemes provided in the NPCI package (Dorie 2016), thus allowing us to know the *true* individual treatment effect. We use 10 replicates of the dataset in this experiment. For each replicate, we divide into three sources, each consists of 249 data points. For each source, we use the first 50 data points for training, the next 100 for testing and the rest 99 for validating. We report the mean and standard error of the evaluation metrics over 10 replicates of the data.

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

—— END ——