# OpenReview forum: "An Adaptive Kernel Approach to Federated Learning of Heterogeneous Causal Effects"
_NeurIPS.cc/2022/Conference — NeurIPS 2022 Accept_

### Official Review · Reviewer_h7Gi · 2022-06-30

**Rating:** 6
**Confidence:** 3
**Soundness:** 3 good
**Presentation:** 2 fair
**Contribution:** 3 good

**Summary:**

The authors propose a new approach called CausalRFF for causal inference (CATE and ATE) based on data from different sources with dissimilar distributions. The approach is built upon Federated Learning such that the risk of privacy leakage can be minimized.

**Questions:**

The paper is generally well-written, but still, I have some questions to ask and want the authors to clarify:

1. In line 200, you write ‘kappa(u^s,u^v)=lambda^{s,v}k(u^s,u^v) if s1\neq s2’. Should it be ‘kappa(u^s,u^v)=lambda^{s,v}k(u^s,u^v) if s\neq v’? If not, how to interpret the formula here? Furthermore, what is the purpose of lambda^{s,v}? Why don’t you introduce lambda^{s,v}k(u^s,u^v) in the case when s1=s2?
2. In line 219, what is L^(s)? Should it be the empirical ELBO from source s? In each source s, should we optimize according to J^(s), and in the federated leverl, we should use Eqn. (13) instead of Eqn. (8)?

**Limitations:**

To my knowledge, this submission has no potential negative societal impact.

**Strengths And Weaknesses:**

Observed data can come from different sources with dissimilar distributions, the authors thus propose a model to learn the distributions separately and adaptively. They use a kernel based model and adapt Random Fourier Features into the model. The authors undergo comprehensive experiments to demonstrate that the proposed approach outperforms other existing methods.

There are some grammatical mistakes. For example, line 202 ‘requires collecting all data points from all source’ should be ‘requires collecting all data points from all sources’, line 188-189, ‘In particular, we propose a kernel-based approach to learn these distribution’ should be ‘In particular, we propose a kernel-based approach to learn these distributions’

---

> ### Author Response · Authors · 2022-08-02
> **Reply to reviewer h7Gi**
>
> Thank you for giving helpful comments to our paper. We would like to reply the reviewer’s comments as follows:
>
> **1. In line 200, you write ‘$\kappa(u^s,u^v)=\lambda^{s,v}k(u^s,u^v)$ if $s_1\neq s_2$’. Should it be ‘$\kappa(u^s,u^v)=\lambda^{s,v}k(u^s,u^v)$ if $s\neq v$’? If not, how to interpret the formula here? Furthermore, what is the purpose of $\lambda^{s,v}$? Why don’t you introduce $\lambda^{s,v}k(u^s,u^v)$ in the case when $s_1=s_2$?**
>
> This is a typo and the correct formula should be $\kappa(u^s,u^v)=\lambda^{s,v}k(u^s,u^v)$ if $s \neq v$ (but not $s1 \neq s2$). Thank you for pointing this out.
>
> For $s=v$, we use $k(u^s,u^v)$ because we know that the two data points $u^s$ and $u^v$ are from the same population (since $s=v$), so the transfer factor should be 1. We use $\lambda^{s,v}$ to adaptively transfer between two different sources $s$ and $v$.
>
> **2. In line 219, what is $L^{(s)}$? Should it be the empirical ELBO from source s? In each source s, should we optimize according to $J^{(s)}$, and in the federated level, we should use Eqn. (13) instead of Eqn. (8)?**
>
> Yes, you are correct. We should optimize Eqn. (13) since it enables federated learning. $L^{(s)}$ is the empirical ELBO from source $s$. We will make it clearer on our revision on this point. Thank you.

---

### Official Review · Reviewer_EKZU · 2022-07-09

**Rating:** 6
**Confidence:** 3
**Soundness:** 2 fair
**Presentation:** 3 good
**Contribution:** 2 fair

**Summary:**

This work is an important step toward privacy-preserving causal inference.
The authors introduce a new approach to causal inference under a specified causal graph from a federated setting with multiple, decentralized, and dissimilarly distributed data sources.
The authors assume the confounders are latent (unobservable) and use use variational inference in the spirit of the variational auto-encoder (VAE) to recover the latent confounders. Further, they propose a kernel-based approach to learn the variational posterior distribution and adapt Random Fourier Features to rewrite the objective function as a summation of local objective functions in each source.


**Questions:**

>1. why assume the causal structure in Figure 1.

I have to say the assumption of $\textit{Causal Structure}$ in Figure 1 seems a rather strong assumption to me.
For one thing, it assumes all the confounders are latent, while in practice the confounders are usually partially observed (some of $Z$ belong to the covariates $X$).

>2. Is it appropriate to use one vertex to denote $X$ in the causal structure

In the causal structure in Figure 1, all the covariates are denoted by one vertex. While in practice (e.g. your real data example), each dimension of $X$ might denote a variable with specific physical meaning, then it is inappropriate to put all the covariates in one vertex. Sorry to stick on the causal structure assumption, I suppose it is quite important since all the content is based on this assumption.

>3. The identification of $Z$

The authors have introduced the identification results in Louizos et al. (2017) in section 3.1. A further question, whether the federated setting would benefit the identification of $Z$, as the assumptions in Louizos et al. (2017) seems very rigid. The authors may refer to the following reference for the identification of latent causal factor in multiple-distributions setting.

Sun X, Wu B, Zheng X, et al. Recovering latent causal factor for generalization to distributional shifts[J]. Advances in Neural Information Processing Systems, 2021, 34: 16846-16859.

>4. the originality of the method

Could the authors explain whether it is the first time to apply the Fourier Features to the optimization process to enable the assertive form of the objective function (equation (13))? As it seems to me a very important innovation of this work.

>5. Experiment

In the real data experiment, whether the ground truth is obtained by raw difference? (since it mentioned IHDP is a randomized study). If so, then I suppose the result is not convincing enough, since it is a very special (degenerative) case with no confounders.


**Limitations:**

The original contribution sees unclear.
The fundamental assumption seems too strong.

**Strengths And Weaknesses:**

> originality:

It seems to me that the problem is original and important (causal inference in federated setting), while the methodology part seems a mixture of existing results (e.g., identifiability of $\boldsymbol{Z}$, solution of $f_c$ in equation (9), the application of Random Fourier Features in equation (10), (12), (13) ). If I misunderstood, please point out which part is original.

> quality&significance:

This is a solid work with contributions to causal inference in federated learning.

> clarity:

The paper is clear and well written.  But it seems overstate the contributions in methodology.

---

> ### Author Response · Authors · 2022-08-02
> **Reply to reviewer EKZU**
>
> Thank you for giving helpful comments to our paper. We would like to reply the reviewer’s comments as follows:
>
> **1. Why assume the causal structure in Figure 1.**
>
> This is a good point. Our work can be extended into two set of confounders: observed confounders and latent confounders. However, the problem is that we would need to find the set of observed confounders. Finding a set of observed confounders is not user-friendly since a non-expert user might not be able to find it. In this case, a solution is to find the observed confounders using a causal discovery algorithm such as PC-algorithm, LiNGAM, NOTEARS, etc. However, it might happen that some observed confounders might not be accurately identified using these algorithms: (i) some non-confounders are learned as confounders, or (ii) some confounders are learned as non-confounders. This might lead to a reduction of the performance of learning causal effects.
>
> In our work, we only consider latent confounders and they are learned with observed proxy variables $X$. This might not be as accurate as separating into two sets of confounders but it would be easier to use in real-life for non-expert users since they don’t have to find the set of observed confounders. Nevertheless, we agree that separating into two sets of observed and latent confounders would improve the performance of the model and this could be a future extension.
>
> **2. Is it appropriate to use one vertex to denote X in the causal structure**
>
> This is a good point. By using one vertex to denote $X$, we are referring to this as a multivariate variable, and there might be causal relations among them. Hence, if we separate $X$ into multiple nodes $X_1, X_2, …X_k$, and if there are no edges among them, then we are assuming that they are independent given $Z$, which is a stronger assumption. Nevertheless, this concern by the reviewer leads to another idea that we might be able to improve the method if we have a causal graph among $X_1, X_2,…,X_k$. This might improve the learning of causal effects since we know more on the structure of $X$, which might be a benefit for learning latent confounders.
>
> **3. Whether the federated setting would benefit the identification of $Z$, as the assumptions in Louizos et al. (2017) seems very rigid. The authors may refer to the following reference for the identification of latent causal factor in multiple-distributions setting: Sun X, Wu B, Zheng X, et al. Recovering latent causal factor for generalization to distributional shifts. Advances in Neural Information Processing Systems, 2021, 34: 16846-16859.**
>
> Federated setting would use more data from multiple sources and thus helpful in learning the two distributions $p(y|w,z)$ and $p(z|x,y,w)$, which is benefit for learning $Z$ and estimating causal effects. The suggested paper is an interesting work on recovering latent causal factor, and it is not in a federated setting, but it is a relevant work which might help improve our work in future. We will refer and discuss this work in our revision. Thank you for the suggestion.
>
> **4. The originality of the method: Could the authors explain whether it is the first time to apply the Fourier Features to the optimization process to enable the assertive form of the objective function (equation (13))? As it seems to me a very important innovation of this work.**
>
> Yes, it is the first time to apply the Random Fourier Features to the optimization process to enable the federated objective function that is transferable among the sources via transfer kernels.
>
> The original use of Random Fourier Features is to accelerate computational time of kernel functions (Rahimi & Recht, 2007).
>
> ***References***
> - Rahimi, A. and Recht, B. Random features for large-scale kernel machines. NeurIPS 2007.
>
> **5. Concern on IHDP dataset**
>
> This is a good observation. The dataset was ‘de-randomized’. The original covariates are confounders and they correspond to collected measurements of the children and their mothers used during a randomized experiment that studied the effect of home visits by specialists on future cognitive test scores.
>
> This dataset was ‘de-randomized’ by removing from the treated set children with non-white mothers; for each unit a treated and a control outcome are then simulated, thus allowing us to know the ‘true’ individual causal effects of the treatment.
>
> We will make this point clearer to avoid confusion in our revision. Thank you.

---

### Official Review · Reviewer_xyps · 2022-07-11

**Rating:** 7
**Confidence:** 4
**Soundness:** 4 excellent
**Presentation:** 3 good
**Contribution:** 4 excellent

**Summary:**

This paper studies the treatment effect estimation from decentralized, heterogeneous data sources that could not be combined together at the individual level. This paper proposes an algorithm based on Random Fourier Features that adaptively learns the similarities among data sources and optimally uses other data sources to learn the treatment effect parameters on each data set. This paper provides minimax lower bounds for the parameters estimated from its proposed algorithm.

**Questions:**

1. Privacy guarantee: Given that the paper is in a federated setting, what can be shared across data sets and which cannot? It will be helpful to elaborate on the privacy constraints (e.g., data cannot be shared at the individual level). It will also be helpful to clarify why the proposed method satisfies the privacy constraints.

2. Effect of heterogeneity on parameter estimation: I really appreciate that this paper provides the minimax lower bounds for the proposed method. However, I think some interpretations of the bounds will make them more accessible to readers. For example, how do the lower bounds vary with the distance between true parameter values (a measure of heterogeneity) across data sets? Some discussion will be helpful.

**Limitations:**

I think this paper can benefit from improving the exposition of its method and main results, especially in Section 4. For example, the authors may consider incorporating the answers to my questions above in the updated version of their paper.

**Strengths And Weaknesses:**

This paper lies at the intersection of causal inference and federated learning, which is an underexplored, but rapidly growing research topic, and has broad applications, especially in healthcare.

This paper accounts for the heterogeneity across data sets, and proposes a novel adaptive algorithm to optimally leverage other data sets to estimate the treatment effect on each source data. Importantly, this paper allows for unobserved confounders and uses variational inference that can identify treatment effects in the presence of unobserved confounders. Note that Vo et al (2021), Xiong et al (2021), and Han et al (2021) assume that all confounders are observed. Therefore, allowing for unobserved confounders is a significant contribution to the literature on federated causal inference.

Moreover, I like the idea of using adaptive kernels and Random Fourier Features to account for heterogeneous data sets and avoid negative transfer, which I think is an important complement to the adaptive method in Han et al (2021).

Overall I think this paper is well-written and I enjoy reading the paper.

---

> ### Author Response · Authors · 2022-08-02
> **Reply to reviewer xyps**
>
> Thank you for your interest in our paper and for giving helpful comments. We would like to reply to your comments as follows:
>
> **1. Privacy guarantee: Given that the paper is in a federated setting, what can be shared across data sets and which cannot? It will be helpful to elaborate on the privacy constraints (e.g., data cannot be shared at the individual level). It will also be helpful to clarify why the proposed method satisfies the privacy constraints.**
>
> In general, we don’t want to share raw individual data among the sources. In our method, we only share the gradients to the server machine which satisfies this constraint. We will clarify this point in our revision. Thank you for the suggestion.
>
> **2. Effect of heterogeneity on parameter estimation: I really appreciate that this paper provides the minimax lower bounds for the proposed method. However, I think some interpretations of the bounds will make them more accessible to readers. For example, how do the lower bounds vary with the distance between true parameter values (a measure of heterogeneity) across data sets? Some discussion will be helpful.**
>
> The minimax lower bounds can be seen as the worst case of the best estimator. It shows that when the adaptive factors are small, the lower bounds are large; this implies that the distance between the true parameter values and the learned parameters are also large under any estimator. In the other case, when the adaptive factors are large, the lower bounds are smaller, which suggests that data from a source would help infer parameters associated with the other sources. We will make it clear in our revision. Thank you for the suggestion.

---

### Official Review · Reviewer_NTzE · 2022-07-11

**Rating:** 6
**Confidence:** 3
**Soundness:** 3 good
**Presentation:** 3 good
**Contribution:** 2 fair

**Summary:**

This paper presents an adaptive kernel approach for learning causal effects from multiple sources. Random Fourier features further extend the kernel to a federated setting.

**Questions:**

Is it possible to scale up the synthetic data experiment regarding clients and data samples?

**Limitations:**

N/A.

**Strengths And Weaknesses:**

Strength:

This paper on federated causal effect is timely and the presented method is reasonable.

Weakness:

1. Whether using a variational auto-encoder (VAE) can preserve the identifiability is unclear because it is known that the common VAE model is unidentifiable [1]. The cited previous work [2] did not claim any identifiability result.

2. The settings regarding the latent variables are not consistent between Lemma 1 and 2. Lemma 1 assumes latent variable but Lemma 2 does not.

3. There are only 5 sources in the experimental setting, which is too few for a federated learning paper.

Reference

[1] Khemakhem, Ilyes et al. “Variational Autoencoders and Nonlinear ICA: A Unifying Framework.” ArXiv abs/1907.04809 (2020).

[2] Louizos, Christos et al. “Causal Effect Inference with Deep Latent-Variable Models.” NIPS (2017).

---

> ### Author Response · Authors · 2022-08-02
> **Reply to reviewer NTzE - Part 1**
>
> Thank you for giving helpful comments to our paper. Here are our comments:
>
> **1. Whether using a variational auto-encoder (VAE) can preserve the identifiability is unclear because it is known that the common VAE model is unidentifiable [1]. The cited previous work [2] did not claim any identifiability result.**
>
> This is a good point. We are not claiming that our model is always identifiable. As stated from line 126,  we are using X (additional variables namely proxies) to infer Z and there are a number of cases that it is identifiable if the following is respected: Z is categorical and X is a Gaussian mixture model (Anandkumar et al. 2014), X includes three independent views of Z (Goodman 1974; Allman et al. 2009; Anandkumar et al. 2012), Z is a multivariate binary and X are noisy functions of Z (Jernite et al. 2013; Arora et al. 2017), to name a few. Because of this, we choose  Z  to be categorical and X is noisy of Z. The paper of Khemakhem  et al. (2020) cited by the reviewer also uses an additional observed variable U for the model to be identifiable. We agree that there is no statistical guarantees on identifiability of our work as it is beyond the scope of the paper. The same is true for Louizos et al. (2017), when the initial model is put forward. It would open an interesting future direction to combine our work with the work by Khemakhem et al. (2020) (the paper cited by the reviewer) to give a statistical guarantee on identifiability.
>
> ***References:***
>
> - Allman, E. S., Matias, C., & Rhodes, J. A. (2009). Identifiability of parameters in latent structure models with many observed variables. The Annals of Statistics, 37(6A), 3099-3132.
> - Anandkumar, A., Ge, R., Hsu, D., Kakade, S. M., & Telgarsky, M. (2014). Tensor decompositions for learning latent variable models. JMLR, 15, 2773-2832.
> - Anandkumar, A., Hsu, D., & Kakade, S. M. (2012). A method of moments for mixture models and hidden Markov models. In Conference on Learning Theory (pp. 33-1). JMLR Workshop and Conference Proceedings.
> - Arora, S., Ge, R., Ma, T., & Risteski, A. (2017). Provable learning of noisy-or networks. In Proceedings of the 49th Annual ACM SIGACT Symposium on Theory of Computing.
> - Goodman, L. A. (1974). Exploratory latent structure analysis using both identifiable and unidentifiable models. Biometrika, 61(2), 215-231.
> - Jernite, Y., Halpern, Y., & Sontag, D. Discovering hidden variables in noisy-or networks using quartet tests. NeurIPS 2013.
> - Khemakhem, I., Kingma, D., Monti, R., & Hyvarinen, A. Variational autoencoders and nonlinear ica: A unifying framework. AISTATS 2020.
> - Louizos, C., Shalit, U., Mooij, J. M., Sontag, D., Zemel, R., & Welling, M. (2017). Causal effect inference with deep latent-variable models. Advances in neural information processing systems, 30.
>
> **2. The settings regarding the latent variables are not consistent between Lemma 1 and 2. Lemma 1 assumes latent variable but Lemma 2 does not.**
>
> Lemma 1 is the minimax lower bound for the learning of two distributions $p(y|w,z)$ and $p(z|x,y,w)$ which contains latent variables. Lemma 2 presents lower bounds for the learning of $p(y|x,w)$ and $p(w|x)$, which have no latent variables. We will make it clearer in our revision. Thank you.
>
> **3. There are only 5 sources in the experimental setting, which is too few for a federated learning paper. Is it possible to scale up the synthetic data experiment regarding clients and data samples?**
>
> This is a good point. Yes, it is possible to scale up the experiment, and we have done extra experiments based on your suggestion as shown in the tables below. We will include them in our revision.
>
> In terms of computational complexity, the number of sources is not a problem since each source would compute its gradient separately. So they can be done in parallel.
>
> In our experiments, we use one machine to run each source separately (since we don’t have multiple machines for the sources). We have scaled up the experiments for 20, 50 and 100 sources as per your suggestion as follows:
>
> ***(Due to limited space, we continue this response below)***

---

> > ### Author Response · Authors · 2022-08-02
> > **Reply to reviewer NTzE - Part 2**
> >
> > ***(We continue our response here)***
> >
> > **Data 1 – The sources have the same distribution**
> >
> > We simulate a dataset of 100 sources, and we use test set from the first 20 sources for evaluating. We set $\Delta=0$ for all sources so that their distribution are the same. The results are as follows:
> >
> > |           | 20 sources $(\sqrt{\epsilon_\text{PEHE}})$ | 50 sources $(\sqrt{\epsilon_\text{PEHE}})$ | 100 sources $(\sqrt{\epsilon_\text{PEHE}})$ | 20 sources $(\epsilon_\text{ATE})$ | 50 sources $(\epsilon_\text{ATE})$ | 100 sources $(\epsilon_\text{ATE})$  |
> > |-----------|:----------:|:----------:|:-----------:|:----------:|:----------:|:-----------:|
> > | BART      | 3.43$\pm$0.03 | 3.40$\pm$0.01 | 3.32$\pm$0.01  | 1.35$\pm$0.06 | 1.31$\pm$0.02 | 1.28$\pm$0.01  |
> > | X-Learner | 2.98$\pm$0.01 | 2.93$\pm$0.01 | 2.90$\pm$0.01  | **0.16$\pm$0.02** | **0.12$\pm$0.02** | **0.13$\pm$0.02**  |
> > | R-Learner | 2.95$\pm$0.01 | 2.92$\pm$0.01 | 2.90$\pm$0.01  | **0.07$\pm$0.01** | **0.10$\pm$0.02** | **0.10$\pm$0.02**  |
> > | OthoRF    | 3.38$\pm$0.03 | 3.32$\pm$0.01 | 3.24$\pm$0.01  | 1.17$\pm$0.06 | 1.10$\pm$0.02 | 1.01$\pm$0.02  |
> > | TARNet    | 3.80$\pm$0.03 | 3.65$\pm$0.01 | 3.27$\pm$0.01  | 1.12$\pm$0.02 | 1.04$\pm$0.01 | 0.93$\pm$0.01  |
> > | CFR Wass  | 3.72$\pm$0.02 | 3.61$\pm$0.01 | 3.19$\pm$0.01  | 1.07$\pm$0.02 | 0.99$\pm$0.01 | 0.87$\pm$0.01  |
> > | CFR MMD   | 3.73$\pm$0.02 | 3.61$\pm$0.01 | 3.20$\pm$0.01  | 1.05$\pm$0.02 | 0.98$\pm$0.01 | 0.87$\pm$0.01  |
> > | CEVAE     | **2.26$\pm$0.01** | 2.21$\pm$0.01 | **1.98$\pm$0.01**  | **0.19$\pm$0.03** | **0.17$\pm$0.01** | 0.17$\pm$0.01  |
> > | FedCI     | **2.24$\pm$0.02** | **2.19$\pm$0.01** | **1.90$\pm$0.01**  | 0.23$\pm$0.04 | 0.21$\pm$0.01 | 0.19$\pm$0.01  |
> > | CausalRFF | **1.58$\pm$0.05** | **1.55$\pm$0.01** | **1.51$\pm$0.01**  | 0.28$\pm$0.04 | 0.21$\pm$0.02 | **0.16$\pm$0.02**  |
> >
> > The table shows that the proposed method achieves competitive results in estimating ATE and CATE when the sources have the same distribution.
> >
> > **Data 2 – The sources have different distributions**
> >
> > We simulate a dataset of 100 sources, and we use test set from the first 20 sources for evaluating. We draw uniformly the discrepancy factor $\Delta \sim Uniform[0,8]$ for each source so that their distributions are different. The results are as follows:
> >
> > |           | 20 sources $(\sqrt{\epsilon_\text{PEHE}})$ | 50 sources $(\sqrt{\epsilon_\text{PEHE}})$ | 100 sources $(\sqrt{\epsilon_\text{PEHE}})$ | 20 sources $(\epsilon_\text{ATE})$ | 50 sources $(\epsilon_\text{ATE})$ | 100 sources $(\epsilon_\text{ATE})$  |
> > |-----------|:----------:|:----------:|:-----------:|:----------:|:----------:|:-----------:|
> > | BART      | 3.43$\pm$0.03 | 3.47$\pm$0.01 | 3.53$\pm$0.01  | 1.35$\pm$0.06 | 1.47$\pm$0.02 | 1.51$\pm$0.01  |
> > | X-Learner | 3.30$\pm$0.04 | 3.23$\pm$0.01 | 3.19$\pm$0.01  | **1.14$\pm$0.08** | 1.20$\pm$0.02 | 1.20$\pm$0.02  |
> > | R-Learner | **3.18$\pm$0.03** | **3.11$\pm$0.01** | **3.08$\pm$0.01**  | **0.88$\pm$0.07** | **0.88$\pm$0.02** | **0.86$\pm$0.01**  |
> > | OthoRF    | 3.38$\pm$0.03 | 3.39$\pm$0.01 | 3.36$\pm$0.01  | 1.22$\pm$0.07 | 1.24$\pm$0.02 | 1.25$\pm$0.01  |
> > | TARNet    | 5.61$\pm$0.04 | 5.63$\pm$0.02 | 5.71$\pm$0.02  | 2.71$\pm$0.06 | 2.75$\pm$0.02 | 2.80$\pm$0.02  |
> > | CFR Wass  | 5.35$\pm$0.05 | 5.37$\pm$0.02 | 5.46$\pm$0.02  | 2.68$\pm$0.05 | 2.69$\pm$0.02 | 2.71$\pm$0.02  |
> > | CFR MMD   | 5.38$\pm$0.05 | 5.40$\pm$0.02 | 5.51$\pm$0.02  | 2.70$\pm$0.05 | 2.68$\pm$0.02 | 2.72$\pm$0.02  |
> > | CEVAE     | 3.37$\pm$0.04 | 3.35$\pm$0.02 | 3.33$\pm$0.01  | 1.20$\pm$0.06 | 1.21$\pm$0.02 | **1.19$\pm$0.01**  |
> > | FedCI     | **3.20$\pm$0.03** | **3.18$\pm$0.02** | **2.98$\pm$0.01**  | 1.21$\pm$0.07 | **1.19$\pm$0.01** | **1.19$\pm$0.01**  |
> > | CausalRFF | **1.75$\pm$0.03** | **1.65$\pm$0.03** | **1.62$\pm$0.01**  | **0.24$\pm$0.04** | **0.19$\pm$0.14** | **0.15$\pm$0.01**  |
> >
> > The table shows that the proposed method outperforms the baselines when the sources have different distributions.
> >
> > These results are consistent with our discussions in the paper. We will include these results in our revision. Thank you again for the suggestion.

---

> > > ### Comment · Reviewer_NTzE · 2022-08-03
> > > **Reply to Authors**
> > >
> > > Thanks for replying and addressing my concerns. I, therefore, raise my score.
> > >
> > > The discussion on identifiability is helpful. Softing the claim and including the additional discussion can improve the paper.

---

> > > > ### Author Response · Authors · 2022-08-04
> > > > **Reply to reviewer NTzE's acknowledgment**
> > > >
> > > > Thank you so much for the reply and raising your score to 6.

---

### Meta-Review · Area_Chair_V6cc · 2022-08-25

**Recommendation:** Accept
**Confidence:** Certain

**Metareview:**

All reviewers found the problem setting, which spans federated learning and causal estimation, important and well motivated, and felt the paper made a solid technical contribution in this space. The reviewers raised questions about topics such as identifiability, the nature of the privacy guarantees, and some technical details. The author responses addressed these questions adequately; one reviewer raised their score. Overall, no major weaknesses were raised. Congratulations to the authors on a solid and well received paper.


**Award:**

No

---

### Decision · Program_Chairs · 2022-09-14

Accept